



# A versatile spaceborne architecture for immediate monitoring of the global methane pledge

Yuchen Wang[1], Xvli Guo[1], Yajie Huo[1], Mengying Li[2], Yuqing Pan[1*], Shaocai Yu[2*], Alexander Baklanov[3], Daniel Rosenfeld[4], John H. Seinfeld[5], and Pengfei Li[1*]

[1]College of Science and Technology, Hebei Agricultural University, Baoding, Hebei 071000, P.R. China
[2]Research Center for Air Pollution and Health; Key Laboratory of Environmental Remediation and Ecological Health, Ministry of Education, College of Environment and Resource Sciences, Zhejiang University, Hangzhou, Zhejiang 310058, P.R. China
[3]Science and Innovation Department, World Meteorological Organization (WMO), Geneva, Switzerland
[4]Institute of Earth Science, The Hebrew University of Jerusalem, Jerusalem, Israel
[5]Division of Chemistry and Chemical Engineering, California Institute of Technology, Pasadena, CA 91125, USA

*Correspondence to: Pengfei Li (lpf_zju@163.com)

Shaocai Yu (shaocaiyu@zju.edu.cn)

Yuqing Pan (panyuqing@hebau.edu.cn)

*Submitted to*

***Atmospheric Chemistry and Physics***





**Abstract.**
The global methane pledge paves a fresh, critical way toward Carbon Neutrality. However, it remains largely invisible and
highly controversial due to the fact that planet-scale and plant-level methane retrievals have rarely been coordinated. This has
never been more essential within a narrow window to reach the Paris target. Here we present a versatile spaceborne architecture
to address this issue. Using this framework, we patrol the world, like the United States, China, the Middle East, and North
Africa, and simultaneously uncover methane-abundant regions and plumes. These include new super-emitters, potential
leakages, and unprecedented multiple plumes in a single source. More importantly, this framework is shown to challenge
official emission reports that possibly mislead estimates from global, regional, to site scales, particularly by missing super-
emitters. We reveal that this framework can enable ready-made satellites to initiate monitoring of the global methane pledge
immediately and is also versatile for upcoming stereoscopic measurements and artificial intelligence techniques.





## 1. Introduction

Global methane pledges finalized at the COP26 (the 26th United Nations Climate Change Conference of the Parties) have been never more ambitious (Schellnhuber et al., 2016; Schurer et al., 2018; United Nations, 2021). More than 100 countries have promised 30% methane emission reductions by 2030. Also, energy giants (e.g., Shell and BP) have committed to clear targets of methane mitigation. Such pledges have never been more essential within a narrow window (< ten years) to reach the Paris target. The scientific context is that atmospheric methane is a powerful greenhouse gas second only to carbon dioxide ($CO_2$), trapping ~ 80 times more heat than the same amount of $CO_2$ (per molecule) over a 20-year time horizon (Etminan et al., 2016; Saunois et al., 2016, 2020). Worse still, it is thought to rise since 2007 (Mikaloff and Hinrich, 2019), surge since 2014 (Nisbet et al., 2019), and set another record in 2021 (National Oceanic and Atmospheric Administration, 2022). Fortunately, methane is short-lived (~ ten years) (K. et al., 2013) and can be reduced in half using existing technologies (Ocko et al., 2021).

However, a classic dilemma emerges, dimming the hopes of scientists and policymakes (Masood and Tollefson, 2021). That is, on the eve of the Paris target, those targets and emissions remain largely invisible worldwide and thus hinder effective mitigation. The main issue is the Paris framework relies on countries or corporate giants to report emissions (A. et al., 2018, 2015; Ganesan et al., 2019). Moreover, the reports are based on indirect statistics, such as O&G inventories, rather than direct measurements(Deng et al., 2022). This leads to a broad consensus that prominent discrepancies exist between the reports. For example, field campaigns nearly double official claims of methane emissions in the United States by correcting leak detection(A. et al., 2018).

To this end, widespread super-emitters present a unique opportunity worldwide (Duren et al., 2019; Pandey et al., 2019; T. et al., 2022; Zavala-Araiza et al., 2015, 2017). They are typically responsible for the underestimates of methane emissions (Alvarez et al., 2018; Duren et al., 2019; Itziar et al., 2021; T. et al., 2022; Thompson et al., 2016). Moreover, there is increasing evidence that methane emissions follow a heavy-tailed distribution (Duren et al., 2019; Frankenberg et al., 2016; T. et al., 2022), for which relatively small number of sources (so-called super-emitters) can account for a disproportionately large share of total emissions. In contrast to area sources (e.g., cities), super-emitters are typically coal mines, wells, gathering stations, storage tanks, pipelines, and flares, with even less than dozens of metres in diameter but high-concentrated methane plumes (Allen et al., 2013; Miller et al., 2019; Subramanian et al., 2015; Varon et al., 2019). We thus anticipate that significant emission mitigation could be achieved by deploying well-designed systems to identify methane super-emitters. For instance, in support of the Paris agreement, the 17th World Meteorological Congress (2015) requested an Integrated Global Greenhouse Gas Information System (IG3IS) that aimed to develop a measurement framework for methane emission reductions (Phil DeCola and WMO Secretariat, 2017).

To date, a large body of field measurements (e.g., in situ and aircraft surveys) between 2012 and 2020 has been designed for methane super-emitters. Despite this, they are spatially confined (e.g., regionally) and temporally infrequent (e.g., a few weeks), incapable of exploring global methane super-emitters (A. et al., 2018; Conley et al., 2016; Duren et al., 2019; Marchese et al.,





2015; Nisbet et al., 2020; Smith et al., 2017; Thompson et al., 2016; Thorpe et al., 2016). Today, substantial advances have
been made towards detecting and quantifying methane super-emitters from space (Cusworth et al., 2019; Hu et al., 2018; Itziar
et al., 2022; Jacob et al., 2016; Pandey et al., 2019; Thompson et al., 2016). Such advances, however, have rarely been expanded
to measure the global methane pledge because large-scale swath and high-resolution sampling have not been coordinated. First,
global methane monitoring has become possible. A flagship satellite mission is the TROPOspheric Monitoring Instrument
(TROPOMI) onboard the Copernicus Sentinel-5 Precursor satellite (T. et al., 2022; Veefkind et al., 2012). It can offer daily
global insights for methane column concentrations, with a large swath width of ~ 2600 km, a moderate resolution of $7.0 \times 5.5$
km$^2$ (since August 2019), and high signal-to-noise ratios. However, its relatively coarse spatial sampling still limits its
application to detect methane super-emitters (T. et al., 2022). Second, next-generation satellite missions, pioneered by the
GHGSat constellation (three satellites at the moment), emerge for mapping methane super-emitters (Cusworth et al., 2019),
with a narrow swath (e.g., ~ 12 km) but a ground-breaking high-resolution spatial sampling (e.g., 25 ~ 50 m)(Jervis et al., 2021;
Varon et al., 2020). Complementary to the GHGSat constellation, satellite-based hyperspectral imager spectrometers, such as
PRISMA, Gaofen-5, ZY1, Sentinel-2, and Worldview-3, have shown great potentials (Guanter et al., 2021; Itziar et al., 2021;
Sánchez-García et al., 2021; Varon et al., 2021). They can resolve methane enhancements and attribute them to specific
infrastructures via similar narrow swath and high-resolution sampling (e.g., 30 m). Note that regions those satellites usually
gazed at are originally well-known home to methane super-emitters. Narrow swath coverage thus remains a crucial limitation
for global surveys of methane super-emitters. Collectively, existing satellite missions still lack both global vision and keen
insight and thus cannot sustain the global methane pledge.
Here we present a multi-tiered, space-based framework for global-scale and high-resolution methane retrievals. The key is that
ready-made satellite missions alone have the potential to initiate immediate monitoring of the global methane pledge. Using
this framework, we patrol the world, with an experimental focus on China, the United States, Iraq, Kuwait, and Algeria, and
reveal both region-scale hotspots and plant-level super-emitters. We can even gaze at a single source to map multiple plumes
and inspect possible methane leakages. These results can challenge national reports that possibly miss unexpected super-
emitters or mislead emission magnitude. On the eve of the Paris target, at least while a global methane monitoring network is
not in place, this multi-tiered satellite constellation presented in this study has important implications for measuring global
methane pledges. Further information on methane retrievals and emission estimates, as well as uncertainty analysis, are shown
in Materials and Methods.
**2. Materials and Methods**
**2.1 Multi-tiered satellite constellation**
The multi-tiered satellite constellation was designed to reconcile global-scale and high-resolution methane monitoring. First,
TROPOMI offered a unique potential for global methane monitoring, depending on its large-scale (i.e., 2600 km) swath, daily
revisit time, regional footprint (i.e., $5.5 \times 7$ km$^2$ since August 2019), and sounding precision and accuracy (i.e., < 1 %)





(Veefkind et al., 2012). Approximately, TROPOMI observed a full swath per second, which resulted in ∼ 216 spectra per
second. This instrument comprised two spectrometer modules, the first involving near-infrared (NIR) spectral channels, and
the second dedicated to the shortwave-infrared (SWIR) spectral channel. The NIR and SWIR channels were equipped with
spectral resolutions of 0.38 and 0.25 nm and spectral sampling ratios of 2.8 and 2.5, respectively. Since the NIR and SWIR
detectors are incorporated in different instrument modules, the NIR spectra will be co-registered with the SWIR spectra before
performing methane retrievals. The methane total column-averaged dry-air mole fraction ($XCH_4$) is retrieved from near-
infrared (NIR) (757 ∼ 774 nm) and shortwave-infrared (SWIR) (2305 ∼ 2385 nm) spectral measurements for sunlight
backscattered by Earth's surface and atmosphere (Hu et al., 2018). In this study, only high-quality measurements, retrieved
under cloud-free and low aerosol load conditions, were used. These measurements were filtered, in addition, for solar zenith
angle (< 70°), low viewing zenith angle (< 60°), and smooth topography ( the surface elevation of < 80 m within 5 km radius)
as described in Hu et al. (28) (Hu et al., 2018).
Hyperspectral satellite missions severed as the second tier, responsible for mapping localized methane super-emitters
depending on their unprecedented resolution (i.e., 3m ∼ 50m). Therein PRISMA, as an open-access representative, was
specifically suitable for this work. It can image the solar radiation reflected by the Earth's surface and atmosphere via hundreds
of spectral channels between the visible and SWIR spectrum (∼ 400 ∼ 2500 nm). Measurements in the SWIR spectrum from
2000 to 2500 nm sampled absorption features from water vapor, carbon dioxide, and methane. Therein the 2100 nm and 2450
nm windows were especially sensitive to methane. Furthermore, the signal-to-noise ratio was reported to be about 100 in the
SWIR for a relatively dark vegetation pixel and increased to above 200 for bright soil surfaces in oil and gas extraction sites.
More importantly, it covered areas of $30 \times 30$ $km^2$ with a 30 m spatial sampling.
We collected dozens of daily measurements from the multi-tiered satellite constellation. These measurements experimentally
mapped regional methane hotspots and localized methane super-emitters across the United States, China, the Middle East (Iraq
and Kuwait), and North Africa (Algeria). The acquisitions were mostly taken between April 2020 and January 2022.
**2.2 Multi-tiered methane retrievals**
In the first tier of our framework, we employed the operational methane products via TROPOMI onboard the Sentinel 5 satellite.
The target product was the column-averaged dry-air volume mixing ratio of methane ($XCH_4$), which will be retrieved
simultaneously with scattering properties of the atmosphere. The operational retrieval algorithm is based on RemoTeC (Butz
et al., 2009; Hasekamp and Butz, 2008), which was originally developed for $CO_2$ and methane retrievals from GOSAT
observations (Butz et al., 2011). It attempted to fit spectra observed by the TROPOMI-based NIR and SWIR channels. Its
sensitivities to atmospheric scattering properties, atmospheric input data, and instrument calibration errors had been
extensively evaluated(Sha et al., 2021; Verhoelst et al., 2021). As a result, the operational products were proved to be critically
stable, with a convergence rate of 99%, and high significance by comparisons with both satellite-based (e.g., GOSAT) and
ground-based (e.g., TCCON) measurements. The required accuracy and precision of < 1 % for the $XCH_4$ product were met
for clear-sky measurements over land surfaces and after appropriate filtering of difficult scenes. Moreover, the forward model





error was less than 1 % for about 95 % of the valid retrievals. Model errors in the input profile of water did not influence the
retrieval outcome noticeably. The methane product is expected to meet the requirements if errors in input profiles of pressure
and temperature remain below 0.3% and 2 K, respectively. Of all instrument calibration errors, the retrieval results were the
most sensitive to an error in the instrument spectral response function of the shortwave infrared channel.
To achieve long-term (i.e., one year) methane retrievals, we oversampled the TROPOMI data at $5 \times 5$ km$^2$ resolution following
Sun et al. (2018) (Sun et al., 2018) where the full spatial footprint of the observation was taken into account by attributing the
observed value to grid cells weighted by the spatial overlap of the observation with those grid cells.
In the second tier of our framework, we applied the matched-filter algorithm to calculate per-pixel methane enhancements with
respect to background levels based on the SWIR sample spectrum (i.e., the 2100 - 2450 nm window) onboard the PRISMA
(Foote et al., 2020; Guanter et al., 2021; Itziar et al., 2021). In theory, the retrieval method can depend on physically-based or
data-driven algorithms. The former aims to explicitly resolve the radiative transfer between the surface, the atmosphere, and
the hyperspectral spectrometers. A key representative is the family of differential optical absorption spectroscopy (DOAS)
methods (Cusworth et al., 2019, 2020, 2021b, 2021a). The latter seeks a methane absorption spectrum across a hyperspectral
image using statistical methods. It is commonly based on the matched-filter and the singular vector decomposition concepts.
These methods are both widely applied and evaluated, especially onboard satellite (e.g., PRISMA, GF-5, and ZY-1) and
airborne (e.g., AVIRIS and AVIRIS-NG) platforms (Cusworth et al., 2020; Foote et al., 2020; Guanter et al., 2021; Itziar et
al., 2021; Thompson et al., 2016; Thorpe et al., 2016).
In this study, the data-driven retrieval based on the matched-filter concept was used. The main reason was that it could
implicitly account for potential radiometric and spectral errors in satellite-based imaging spectroscopy. For instance, vertical
striping was prevalent in hyperspectral measurements due to detector inhomogeneity, thus substantially degrading methane
retrievals. The matched-filter algorithm focused on the per-pixel columns and thus tackled this issue in principle. Besides, the
physically-based method had to consider background concentrations that were difficult to determine around the super-emitters.
In contrast, the data-driven method was independent of background levels and can directly seek methane enhancements. Finally,
the data-driven method generally had a substantially superior computational efficiency compared to the physically-based
method.
The matched-filter retrieval used here was similar to the one used by Thompson et al. (2016) (Thompson et al., 2016) for the
Hyperion imaging spectrometer onboard the EO-1 satellite. The calculation processes of methane enhancements ($\Delta$**XCH₄,**
**ppb**) were as follows.
$\Delta \mathbf{XCH_4}(\vec{\mathbf{x}}) = \frac{(\vec{\mathbf{x}} - \vec{\mu})^T \Sigma^{-1} \vec{\mathbf{t}}}{\vec{\mathbf{t}}^T \Sigma^{-1} \vec{\mathbf{t}}}$ (Eq. 1).
The $\vec{x}$ denoted the spectrum under analysis. The $\vec{\mu}$ and $\Sigma$ represented the mean background radiance and corresponding
covariance, respectively. They were calculated based on per-column spectrums in order to consider different responses of
across-track detectors to radiance. The $\vec{t}$ was the target spectrum that reflected the background radiance enhanced by the
methane plume. It was generated by the elementwise multiplication of $\vec{\mu}$ and $\vec{k}$, This implicit parameter $\vec{k}$ represented a unit





methane absorption spectrum derived from a look-up table simulated by the MODTRAN radiative transfer model. Similarly,
the spectral convolution was also performed on a per-column basis.
In theory, methane enhancements detected in spectrometers generally exhibit sparsity, especially over low albedo surfaces.
We thus accounted for such non-specificity effects to improve the basic version of the matched-filter model. A major measure
to compensate for the albedo effect was to scale the target spectrum $\vec{t}$ by the pixel-specific albedo factor due to the fact that
the Beer–Lambert absorption law depended on the initial radiance in the absence of the absorber. Here the pixel-specific scalar
$f$ was calculated based on the spectral average $\vec{\mu}$ and the analysis spectrum $\vec{x}$ as follows:
$f = \frac{\vec{x}^T\mu}{\mu^T\mu}$. (Eq. 2)
This solution made $\Delta\mathbf{XCH_4}$ normalized by the albedo term, which was similar to the per-pixel normalization in previous
hyperspectral analysis (Kraut et al., 2005).
The premise to launch the matched-filter algorithm was the accurate knowledge of the response of the instrument spectra to
the methane absorption nature. To this end, the objective was to gain the best fit between the simulated and reference spectra.
An initial step was thus conducted to update the spectral calibration for the channels within the 2100 - 2400 nm window, in
which the channel wavelength centre and width were updated for each across-track position in each scene. Other details are
illustrated in previous attempts (Foote et al., 2020; Guanter et al., 2021; Itziar et al., 2022).
**2.3 Multi-tiered attribution of methane hotspots and plumes**
In the first tier of our framework, we filtered the TROPOMI-based methane retrievals to identify region-scale hotspots. In each
scene, we focused on the grids with anomalous values that were noticeably higher than the average. In this study, we used a
Boolean mask to define the pixels that were affected by the methane emissions (Pandey et al., 2019). Although automatic
threshold and algorithm might result in more consistent and flexible identifications, no satisfactory set of criteria was found
that could be applied for this study. This was mainly because, in localized regions, the methane budgets responded to the
changes in not only the super-emitters but also the background. Assisted by artificial intelligence techniques in the future
(Ouerghi et al., 2021; Paoletti et al., 2018; Yang et al., 2018; Yu et al., 2017; Zhang et al., 2018), our framework can derive a
global, operational, and open-access methane monitoring network. As expected, multiple hotspots of interest result, and here
we focused on those in the United States, China, Iraq, Kuwait, and Algeria.
In the second tier of our framework, we applied visual inspection to detect plumes using the PRISMA-based methane retrievals
(Itziar et al., 2021; Martin et al., 2018; T. et al., 2022; Varon et al., 2020). To date, it was still challenging to distinguish
methane plumes in hyperspectral images using full physically-based algorithms. The main cause was potential methane
retrieval artifacts from hyperspectral satellites that were spatially correlated to surface features. Specifically, we manually
searched for methane enhancement pixels with gas-plume-like shapes, i.e., high methane enhancements progressively
decreased downwind. The resulting pixels were subsequently compared to the spectral radiance data at the 2300 nm absorption
feature sensitive to low surface albedos. In this way, the fake positives due to specific surface features were prevented. On this





basis, the candidate pixels were overlaid over simultaneous (i.e., hourly) wind fields and high-resolution imageries in
individual scenes. They would be considered to be true plumes if they roughly aligned with simultaneous wind direction and
originated from explicit infrastructures. Here the high-resolution satellite imageries were taken from the Google Map. The
hourly wind field data came from the ERA5 reanalysis dataset produced by the European Centre for Medium-Range Weather
Forecasts (ECMWF) (Hersbach et al., 2020; Hoffmann et al., 2019). Finally, we manually drew polygons to mask such
resulting plumes out. As preparation for plume emission quantification, we removed the background using the threshold of the
median values of the scenes.
These satellite imageries allowed us to categorize methane plumes within narrow spatial scales between 50 to 500 m$^2$, such as
O&G extraction platforms, storage tanks, and compressor stations. They even enabled the attribution of plumes to specific
emission ports in individual sources due to their very high resolution. Furthermore, we could name them based on points of
interest in the Google Map. On this basis, such sources could be visually retrospected via long-term, high-resolution (i.e., 10
m) satellite images from the Sentinel-2 mission (Ehret et al., 2021; Varon et al., 2021). Their key details, like ages and statuses
(e.g., active or inactive), were thus collected reliably. Note that, regarding such information, national reports were typically
credible but inaccessible, particularly in global missions. In addition, it should be highlighted that, on top of considerably high
budgets, like megacities, there must be super-emitters undetectable in our way. Other causes are discussed in uncertainty
analysis in Supplement Information.
**2.4 Multi-tiered quantification of methane emissions**
In our framework, we calculated the total excess mass of methane in kilograms in the detected hotspots (in the first tier) and
plumes (in the second tier) using the so-called integrated mass enhancement (IME) model(Frankenberg et al., 2016; Varon et
al., 2018). To make conservative estimates, we defined the background levels as the 10% of the average methane
concentrations in the TROPOMI-based and PRISMA-based scenes (Figs. 1b ~ 1g)(Frankenberg et al., 2016; Varon et al.,
2018). On this basis, we eliminated the interferences from the background concentrations and calculated IMEs as the methane
masses of the masked hotspots and plumes.
Overall, this method linked the emission rate ($\boldsymbol{Q}$) with the measured IME via the residence time of methane ($\boldsymbol{IME/Q}$). This
residence time relied on an effective wind speed ($\boldsymbol{U_{\text{eff}}}$) and a characteristic plume size ($\boldsymbol{L}$) as follows:
$\boldsymbol{Q} = \dfrac{U_{\text{eff}} \cdot \text{IME}}{L}$. (Eq. 3)
Specifically, the $\boldsymbol{IME}$ and $\boldsymbol{L}$ can be inferred from the observations of the hotspots or plumes. During this process, we carefully
applied a Boolean plume mask that separated the pixels ($\boldsymbol{i}$) with notable signals ($\Delta\boldsymbol{\Omega_i}$) from background pixels and thus defined
the total areas ($\boldsymbol{\Sigma_{i=1}^{N} A_i}$) of the hotspots or plumes. The $\boldsymbol{L}$ was defined as the square root of the total plume areas. Hence, the
$\boldsymbol{IME}$ was calculated as follows:
$\textbf{IME} = \Sigma_{i=1}^{N} \Delta\Omega_i A_i$. (Eq. 4)



In the first tier of our framework, the effective wind speed ($U_{\mathrm{eff}}$) was defined as the 10-m wind speed $U_{10}$ obtained from the
ERA5 reanalysis dataset. According to the detected hotspot, the value at the nearest hour and location were used.
In the second tier of our framework, we applied an ensemble of large eddy simulations (LES) to establish an empirical, linear
relationship between $U_{\mathrm{eff}}$ and the measured 10-m wind speed $U_{10}$ as follows (Fig. S9)
$U_{\mathrm{eff}} = 0.8602 In(U_{10}) + 1.1513.$ (Eq. 5)
The configurations of these simulations, such as spatial resolution and precision, were comparable to our PRISMA data. Other
details in this methodology were described in Varon et al. (2018) (Varon et al., 2018).
We estimated the uncertainties of $Q$ by propagating the random errors in $U_{10}$ and **IME**. This processes were conducted in
previous studies(Cusworth et al., 2019, 2021b; Itziar et al., 2021). As shown in previous findings, the major error source came
from the $U_{10}$ term. Its random distributions typically corresponded to the 50% random error. On this basis, this error was
integrated quadratically with the standard error of the **IME**, the result of which can be treated as the final random error of $Q$.
The intrinsic errors of the IME model were quantified in the following uncertain analysis.

**2.5 Uncertainty Analysis**

The objective of this work was to promote a multi-tiered satellite constellation that can monitoring global methane pledges.
To better understand the performance of our framework, we conducted comprehensive uncertain analysis. Note that the
protocol of the uncertain analysis on our framework originated from previous studies (Itziar et al., 2021; Varon et al., 2020).
Specifically, we required to account for the uncertainties in the TROPOMI-based and PRISMA-based methane retrievals and
subsequent emission estimates. Therein the operational TROPOMI-based methane retrieval products had been evaluated
strictly and proved to be reliable globally (except in low- and high-albedo and snow-covered areas) (Lorente et al., 2021; Sha
et al., 2021). In this work, we thus focused on three main sources of uncertainties, specifically including (1) uncertainties in
the PRISMA-based methane retrievals; (2) uncertainties in the TROPOMI-based methane emission estimates; and (3)
uncertainties in PRISMA-based methane emission estimates. During the analysis for the latter two uncertain sources, we would
further investigate the potential wind impacts on the methane emission estimates. Note that it remained challenging to directly
quantify the uncertainties in the wind fields across our cases due to the lack of measurements. We would thus assess the
variations in the methane emission estimates driven by distinct wind data. From such analysis, we could confirm the reliable
performance of our framework. Details can be found in Supplementary Information.

**3 Results and discussions**

**3.1 Multi-tiered imaging of global methane hotspots and super-emitters**

Figure 1 presents representative sets of methane hotspots and associated super-emitters across the United States, China, the
Middle East (Iraq and Kuwait), and North Africa (Algeria) via our multi-tiered satellite constellation. Each group first clarified



a methane-abundant region and further focused in on explicit super-emitters. Among them, five methane-abundant regions
were captured in Wattenberg (the United States), Yangquan (China), Rumaila (Iraq), Burgan (Kuwait), and Hassi Messaoud
(Algeria) (Fig. 1a and Table S1). These accounted for 4805 ~ 46138 kg/h methane emissions based on our daily first-tiered
(i.e., TROPOMI-based) monitoring. From the perspective of a state-of-the-art global methane emission inventory (i.e.,
EDGARv6.0), such high values ranked among the top 1% regarding emission intensities per unit area ($km^2$) (Fig. S1)(Crippa
et al., 2020). The Rumaila field, for example, was known as the largest oil field in Iraq (in terms of both reserves and yields).
In this work, it was found with a significant methane emission intensity exceeding 45000 kg/h (Fig. 1b). In addition to such
well-known oil fields (Figs. 1c ~ 1f), methane hotspots emerged in developing coal mines, like Yangquan, with comparable
emission levels (> 30000 kg/h) (Fig. 1g).
We attributed these methane enhancements to specific methane plumes via the second-tiered (i.e., PRISMA-based) monitoring
(Figs. 1b1 ~ 1g2). There are substantial variations in the methane plumes' amounts, types, and magnitude, even in a single
methane-abundant region. For instance, in the Burgan field, the second-tiered monitoring detected up to eight methane plumes
in a handful of grids in the first-tiered monitoring (Figs. 1c1 ~ 1c4 and 1d1 ~ 1d4). Such intensive distributions were also
found in previous region-oriented surveys in the Permian basin and California (Duren et al., 2019; Itziar et al., 2021). Together
with high-definition images (Fig. S2), we found that such plumes originated from various sources, such as flares, factories,
and wells. A breakthrough was the capture of two distinctive plumes in an individual methane source with extremely high
emissions (> 10000 kg/h), unprecedented in previous satellite-based exploration and only observable in aircraft surveys (Fig.
1b1). Such precise distinction benefited from the high resolution of the second-tiered monitoring, despite being limited by the
relatively higher detection threshold (~ 300 kg/h) (Guanter et al., 2021). Besides, factories and wells can also emit such evident
plumes (Fig. 1c1 and Figs. 1e1 and 1e2). By comparison, other plumes were typically more diffuse but with comparable
emission magnitude (~ 1000 ~ 7000 kg/h).
Note that the above results represent only snapshots at the overpass moments of the satellites (i.e., TROPOMI and PRISMA)
(Figure 1). Specifically, for a given set (including both a methane-abundant region and associated super-emitters), the overpass
timing of TROPOMI can be nearly concordant with that of PRISMA. The temporal gaps could be frequently controlled within
ten days (e.g., Figs. 1b and 1d), even two days (Figs. 1e, 1f, and 1g). For instance, within only two days (August 18th and 19th,
2021, November 15th and 17th, 2021), our multi-tiered satellite constellation went through the Hassi Messaoud field and the
Yangquan coal mine and provided in-depth views of methane budgets, including methane-abundant regions and their drivers
(Figs. 1e and 1g). Even, in just one day (July 7th, 2021), our multi-tiered satellite constellation not only uncovered methane
enhancements in the Wattenberg field (Fig. 1f) but also tracked them back to explicit methane super-emitters (Figs. 1f1 and
1f2). As expected, if we extended the monitoring window of our framework to years, more methane super-emitters were
subsequently captured (Figs. S3 and S4). Moreover, our framework via multi-tiered satellite constellation paves an in-time
way for routine monitoring of global methane hotspots and associated super-emitters.



### 3.2 Multi-tiered verification of global methane super-emitters

Four unexpected cases occurred in Burgan (Iraq), Hassi Messaoud (Algeria), and Yangquan (China), potentially explainable
if we took mutual verification of the first- and second-tiered monitoring into consideration. First, an anomalous methane plume
was detected in the Burgan field (Fig. 1c4) of high emission magnitude (> 1500 kg/h), notably exceeding typical O&G facilities,
from an elusive source (i.e., no clear source could be attributed) (Fig. S2). The long-term measurements of our multi-tiered
satellite constellation intermittently, rather than accidentally, observed this abnormal plume (Figs. S4). Furthermore, uncertain
analysis (see Materials and Methods) helped confirm this real plume. In particular, the methane plumes were clearly
uncorrelated with the surface brightness from space (Fig. S5). Consequently, the most likely hypothesis for this super-emitter
was methane leakage from gigantic O&G pipelines as shown in the Google Map (Fig. S2).
Second, we observed suspect trails of methane plumes above the storage tanks in the Burgan field (Fig. 1d4). Conceivably, the
technical noise driven by albedo effects bore the brunt, although it was believed to be corrected reliably (See Materials and
Methods). To this end, we applied a multi-spectral retrieval algorithm to eliminate this effect to a large extent. The detailed
illustrations are shown in Materials and Methods (Fig. S6). Consequently, we provided evidence that un-negligible methane
emissions (> 3500 kg/h) may very well be the unique explanation, likely related to fugitive methane leaks from the storage
tanks. This was only seen in previous aircraft-based surveys(Frankenberg et al., 2016). Therefore, our multi-tiered outcomes
indicate even more widespread methane leaks than expected. Note that the multi-spectral retrieval algorithm cannot completely
remove the albedo effects on our framework. As such, our framework could lead to efficient on-site re-inspection on worldwide
and innumerable O&G fields.
Third, our framework detected a new methane super-emitter in the Hassi Messaoud field on December 7, 2021 (Fig. 1e4). By
revisiting historical satellite images in the second-tiered monitoring (Fig. S7), we could confirm that this super-emitter arose
between October 18th and November 12, 2021. These results indicate that monitoring of global methane super-emitters can
attain monthly resolution via current satellite constellation alone. Conceivably, more satellite observations would further close
the time window. Fourth, a distinct methane plume appeared in a coal mine in a mountainous area (Yangquan, China),
exceeding all of the detected O&G super-emitters regarding the emission rate (> 7000 kg/h) (Fig. 1g1).
Figure 2a illustrates that, in our multi-tiered satellite constellation, the extent to which the explicit plumes in the second tier
explained the regional budget detected by the first tier. Overall, the plumes in the former were mostly responsible for large
shares (> 8.2%) of regional budgets in the latter. In the Rumaila, Burgan, and Wattenberg fields, the detected methane plumes
played a more critical role, with contributions up to 53.8 ~ 65.9%. Note that such contribution estimates might occasionally
exceed 100% mainly owing to the inconsistent overpass moments between the first- and second-tier monitoring. By
comparison, the relatively low but still significant contributions in the Hassi Messaoud field (8.2%) and Yangquan coal mine
(35.7%) were partly due to the technical limitation of our framework in detecting methane plumes on top of high background
levels. Collectively, the heavy-tail law of methane plume distributions, early reported for regional O&G fields (like the Permian
basin and California) (Duren et al., 2019; Itziar et al., 2021), possibly applied worldwide.





To further explore such a hypothesis, we extended the temporal sample window of our multi-tiered framework. As expected,
the first tier pictured less vivid methane enhancements mainly due to wind averages (Martin et al., 2018; McLinden et al.,
2016; Sun et al., 2018) (Fig. S3), while the second tier could capture more methane plumes (Fig. S4). This would lead to more
volatile contributions of the plumes to regional budgets, which, however, remained at high levels. This reinforces the above
hypothesis for the widespread occurrence of methane super-emitters.
A regional survey in a California field was considered as the best reference, owing to its utilization of systematic airborne
measurements to detect and quantify methane super-emitters (Duren et al., 2019). They survey reported 1181 methane plumes,
more than 500 times larger than previous aerial studies (Englander et al., 2018), with a median emission intensity of 170 kg/h.
These results were thus used to directly evaluate the outcomes in the second tier (Fig. 3). Even though some regions of interest
in this study were far less famous than the California field, their emission intensities were much higher. Specifically, these
plumes detected by the second-tiered monitoring had emission intensities (1142 ~ 11698 kg/h) that exceeded the median value
in the California field. Satellite-based surveys conducted repeatedly for the Permian basin (one of the top O&G bases
worldwide) from 2019 to 2020 (Fig. 3) as compared to the surveys in the California field, they achieved a much higher number
of strong methane super-emitters, the median emission rates (1850 kg/h) much closer to ours (2888 kg/h). Collectively, this
direct comparison indicates the outstanding strength of our results that, though derived from a small dataset, could be analogous
to abundant outcomes from field campaigns. More importantly, this highlights the urgent need for global monitoring of
'nameless' O&G facilities that possibly emit methane as much as the California field and Permian basin.
**3.3 Multi-tiered challenges of national emission inventories**
These multi-tiered results challenge traditional methane emission inventories (Fig. 4). Here the conventional emission data
was obtained from a state-of-the-art bottom-up emission inventory (i.e., EDGARv6.0) for the year 2018. Consequently, for
the methane hotspots, this inventory was mostly consistent with the present results (-49.9 ~ 91.8%), with a fine average bias
(63.2%). The Hassi Messaoud field in Algeria was a unique exception, where the O&G sector was in rapid development, with
a relatively larger bias (489.2%). By comparison, this inventory significantly undervalued the methane super-emitters (up to
orders of magnitude). This indicates that traditional emission inventories might have acceptable performance for traditional
methane-abundant regions while incapable of tracking methane super-emitters.
First, outdated spatial proxies might explain the large divergence between our plant-based estimates and the EDGARv6.0 (Fig.
1b1 and Fig. S8). Moreover, the EDGARv6.0 was designed for the year 2018, missing the newly established O&G plants with
high methane emissions. Second, in principle, conventional inventories directly missed high emissions caused by abnormal
operations (e.g., equipment failures) (Fig. 1c4 and Fig. S8) such as the O&G blowout shown in on-site surveys (Pandey et al.,
2019). A compromise was downwind measurements, yet insufficiently reliable as shown in previous findings(A. et al., 2018).
In addition, the relatively low bias in the Rumaila and Hassi Messaoud fields might be explained by other causes (Figs. 1b2
and 1e3) such as outdated emission factors. Empirically, a plant-level inventory, once optimized by direct measurements, can




raise total methane emissions by ~ 60%, although source categories vary substantially (A. et al., 2018). Besides, temporal
variability might also explain top-down and bottom-up differences in methane emission estimates. For instance, the peak
emission rate could exceed 40% higher than the average, which might occur in the middle afternoon due to specific processes,
like episodic venting from manual liquid unloading (Vaughn et al., 2018). This aligned with the sampling time of the satellites,
thus biasing bottom-up inventories. Collectively, it is necessary to carefully consider all factors affecting methane emissions,
including emission factor updating and spatiotemporal variations, in order to develop effective strategies for mitigating
methane emissions.
**3.4 Implications for global methane monitoring**
We present a multi-tiered, space-based framework that can harmonize planet-scale and plant-level methane retrievals (Fig. 5).
Using this framework, we patrol the world, with synergistic, proactive detections on the methane-abundant regions and
methane super-emitters across the United States, China, the Middle East (Iraq and Kuwait), and North Africa (Algeria). We
even lock new methane super-emitters, track potential methane leakages from storage tanks, and distinguish multiple methane
plumes in a single source. Such achievements are mostly unprecedented in satellite surveys and only observed in aircraft
campaigns. On this basis, our results challenge national reports that possibly miss unexpected super-emitters or mislead
emission magnitude, partly due to surges of oil and gas (O&G) facilities and widespread abnormal operations.
Our data prove that depending on ready-made satellite missions alone can initiate immediate, proactive monitoring of global
methane pledges, in contrast to existing surveys that have to focus on a priori methane-abundant regions. As such, as the
window for achieving the Paris target is rapidly closing, we will not need to sit back and wait for upcoming space missions,
like MethaneSAT and SBG in the United States, EnMAP in Germany, a new version of GF-5 in China, and, later, the European
Space Agency's CHIME from 2025 to 2030 (Cusworth et al., 2019). In addition, while scientific communities persistently
debates the drivers of the recent methane surge (G. et al., 2014; Nisbet et al., 2019; Turner et al., 2019), the consequences of
our outcomes are clear, not only holding clues but also facilitating mitigation.
It should be noted that the multi-tiered framework is sustainable (Fig. 5). On the one hand, it can harmonize multiple satellites.
The potential representatives include upcoming official missions (e.g., the GF-5) (Itziar et al., 2021), current private
constellations (e.g., the GHGSat series) (Jervis et al., 2021; Varon et al., 2020), and explorable multispectral products (e.g.,
the Worldview-3 and Sentinel-2) (Sánchez-García et al., 2021). On the other hand, the framework is not confined to satellites
and can be expanded by integrating in situ (e.g., Global Atmosphere Watch Programme) (World Meteorological Organization,
2022), aircraft, and unmanned aerial vehicles (UAVs) (Cusworth et al., 2020; Gålfalk et al., 2021; Tuzson et al., 2020).
Particularly, on the basis of our framework, rapid advances in artificial intelligence (AI) techniques are projected to completely
replace manpower to seek faint signals of methane enhancements in Earth's surface, and to significantly optimize data-driven
algorithms of methane emission estimates (Reichstein et al., 2019; Yuan et al., 2020). In principle, subsequent mitigation of





such super-emitters via routine maintenances, leak detections, or emergent repairs can provide effective, efficient, and
economical solutions toward the Paris target (Mayfield et al., 2017).
These outcomes have important ramifications for low- and middle-income countries. World powers, like the United States and
European Union, lead new national methane pledges. They are separately on the way to creating vast operational infrastructures
to monitor ambitious climate goals. Still, huge holes remain in coverage and authority, at least by the middle of this decade.
This situation is especially worse for low- and middle-income countries, where the tight budget dims the hopes for filling up
those holes by 2030, while methane emissions are likely to rise as countries develop. In this context, the present framework
can at once serve as the cost-effective piece of the global methane monitoring network and thus support fair climate
negotiations between countries.







**Fig. 1. Methane hotspots and associated super-emitters across the United States, China, Iraq, Kuwait, and Algeria via**
**the multi-tiered daily satellite constellation.** (**a**) Methane-abundant regions and associated super-emitters are captured by
the TROPOMI and PRISMA, respectively. Their locations are marked by black rectangles and dots. Their names are obtained
from the Google Map, usually being the names of the nearest O&G fields and coal mines. (**b ~ g**) Each group clarifies a
methane-abundant region and explicit super-emitters (b1 ~ b4, c1 ~ c4, d1 ~ d4, e1 ~ e4, f1 ~ f2, and g1 ~ g2). For each super-
emitter (five-pointed stars), the overpass moments of the multi-tiered satellite constellation and the consequent emission
estimate are presented. Its base map is obtained from the ©Google Map. The second color bar for the PRISMA is suitable
for the super-emitters in China, while the first is for other countries.



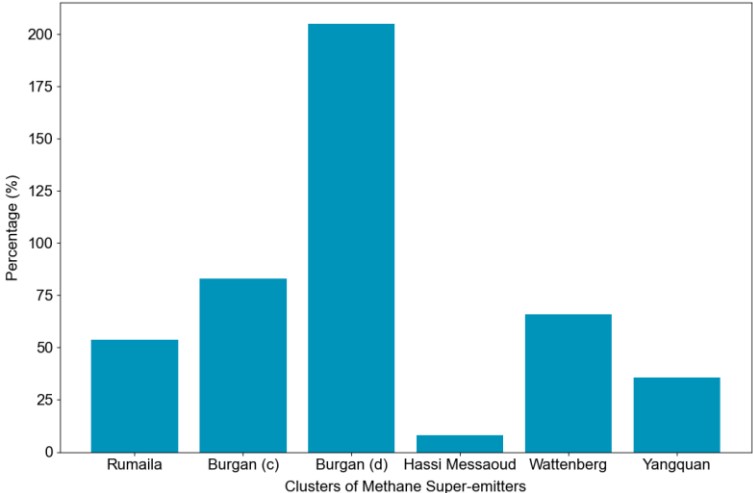


**Fig. 2. High contributions of methane super-emitters to corresponding regional methane budgets.**

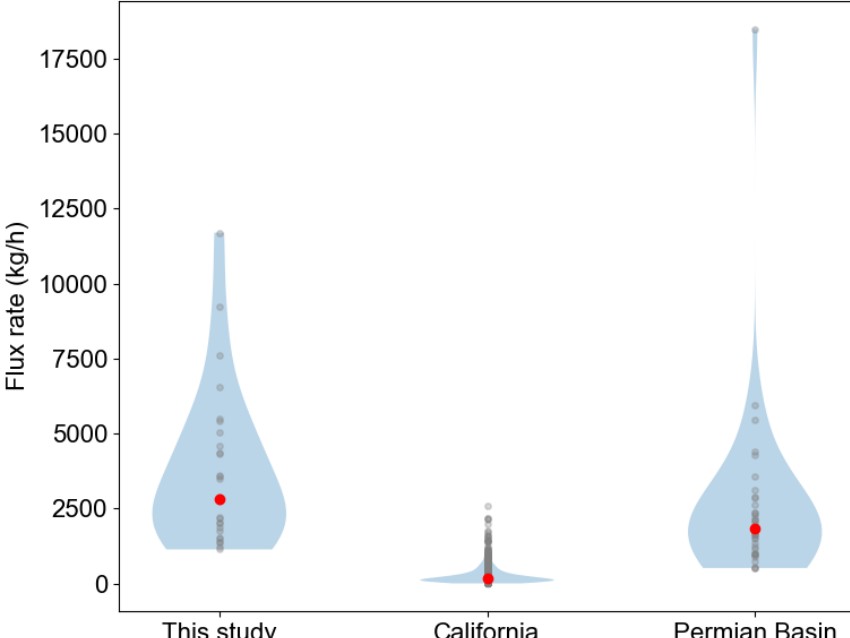


**Fig. 3. Comparison of emission estimates of methane plumes between surveys.** The surveys for the California field and

Permian basin are selected as the references. They report 1181 and 39 methane plumes, while our second-tiered survey attempts

29 plumes. Violin plots show statistical distributions of methane plume emission rates for these surveys. For each survey, the

grey dots refer to the emission rates of the individual plumes and the red dot represents the median value.

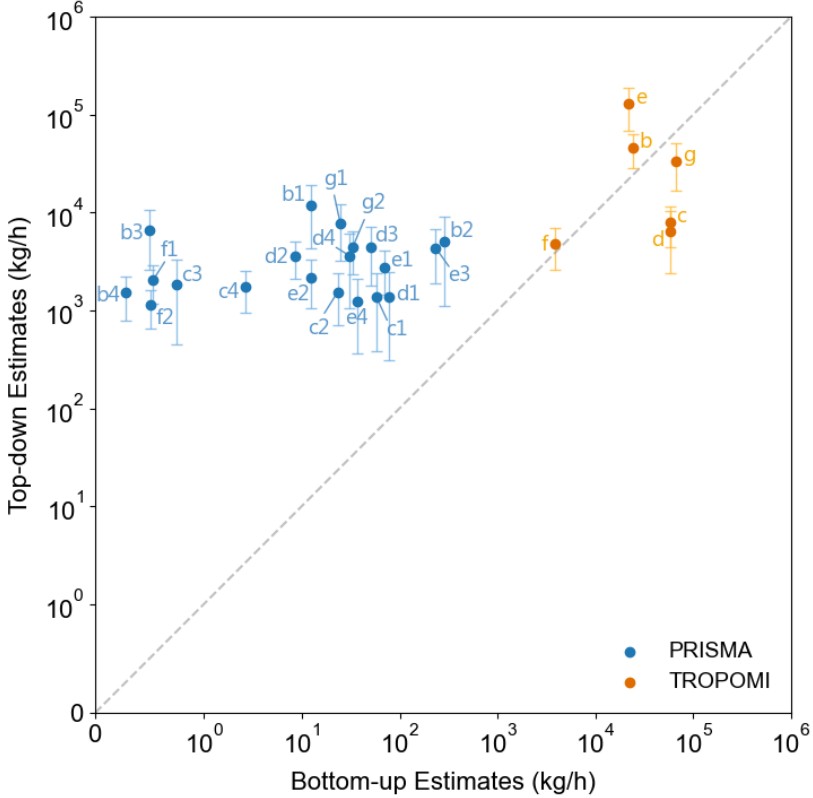

**Fig. 4. Multi-tiered emission estimates versus bottom-up emission inventories.** We first interpolate the bottom-up emission
inventories into the resolution consistent with our multi-tiered results. On this basis, the bottom-up emission rates in the grids
that the detected hotspots and plumes cover are summed up to compare with the results. The detected hotspots (yellow dots)
and plumes (blue dots) correspond to those as shown in Fig. 1. The grey dashed line represents the ratio of the bottom-up
emissions to the top-down ones of 1:1.





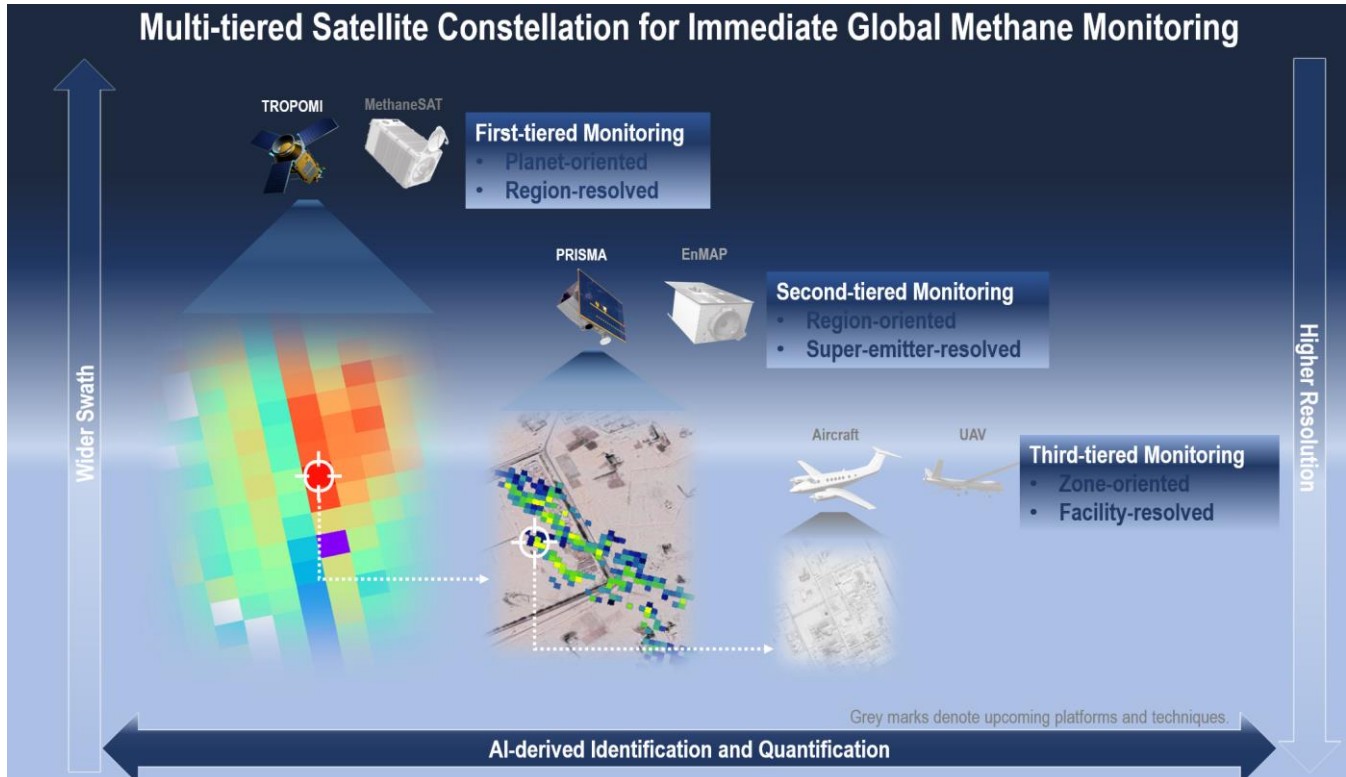


**Fig. 5. Multi-tiered satellite framework for immediate global methane monitoring.** This framework harmonizes global-scale and high-resolution methane retrievals, with a dual focus on mapping region-scale and plant-level drivers. In this work, the framework reconciles the spacious swath of TROPOMI (i.e., ~ 2600 km) with the high resolution of PRISMA (i.e., 30 × 30 m$^2$), in contrast to conventional satellite-based surveys that were of either insufficient samplings or narrow views. Looking forward, developments of Earth's monitoring platforms (e.g., satellites, aircrafts, and unmanned drones) and artificial intelligence will continue to strengthen the performance of methane plume retrievals and emission estimates. On eve of the Paris target, at least while a super methane satellite with spacious swath, high resolution, and agile analysis is not in place, our multi-tiered satellite constellation has important implications for measuring global methane pledges. The appearances of the TROPOMI, MethaneSAT, PRISMA, and EnMAP are obtained from http://www.tropomi.eu/, https://www.methanesat.org, https://www.asi.it/en/earth-science/prisma/, and https://www.enmap.org/, respectively. The methane maps from the TROPOMI and PRISMA refer to the results in Figs. 1e and 1b1. The grey marks indicate upcoming platforms (i.e., MethaneSAT and EnMAP) and techniques (e.g., AI techniques that can optimize the identification and quantification of methane super-emitters).

440



**Data availability.**

The operational TROPOMI product is available at https://scihub.copernicus.eu/, https://www.temis.nl/emissions/data.php. The PRISMA data are publicly available to registered users at https://prisma.asi.it/. The WRF-CHEM model code is available at https://ruc.noaa.gov/wrf/wrf-chem/. All Sentinel-2 satellite data are publicly available through the Copernicus Open Access Hub (https://scihub.copernicus.eu/). The HITRAN line spectra is publicly available through the HITRANonline database (https://hitran.org/). The ERA5 data come from https://www.ecmwf.int/en/forecasts/datasets/reanalysis-datasets/era5. The EDGARv6.0 dataset comes from https://edgar.jrc.ec.europa.eu/gallery?release=v60ghg&substance=CH4§or=TOTALS.

**Code availability.**

The codes are available upon request to corresponding author.

**Supplementary information.**

Supplementary information accompanies this paper.

**Author contributions.**

P. L. designed this study and wrote the manuscript. P. L. and Y. W. developed the retrieval algorithm. P. L., Y. W., X. G., Y. H., and Y. P. performed the data analysis. S. Y., A.B., D. R., and J. H. S. contributed to the manuscript.

**Competing interests.**

The authors declare no competing interests.

**Acknowledgements.**

We thank ESA and the S-5P/TROPOMI team for the great work on initiating and realizing TROPOMI data. We also thank the Italian Space Agency for the great work on the PRISMA data. This study is supported by National Natural Science Foundation of China (No. 22006030, 22076172, 21577126 and 41561144004), Science and Technology Program of Hebei Province (22343702D), Hebei Youth Top Fund (BJ2020032), Research Foundation of Education Bureau of Hebei (QN2019184), Basic Scientific Research Foundation of Hebei (KY2021024), Initiation Fund of Hebei Agricultural University (412201904 and YJ201833), the Department of Science and Technology of China (No. 2016YFC0202702, 2018YFC0213506 and 2018YFC0213503), and National Research Program for Key Issues in Air Pollution Control in China (No. DQGG0107).





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

Gålfalk, M., Nilsson Påledal, S. and Bastviken, D.: Sensitive Drone Mapping of Methane Emissions without the Need for
Supplementary Ground-Based Measurements, ACS Earth Sp. Chem., 5(10), 2668–2676,
doi:10.1021/acsearthspacechem.1c00106, 2021.
Ganesan, A. L., Schwietzke, S., Poulter, B., Arnold, T., Lan, X., Rigby, M., Vogel, F. R., van der Werf, G. R., Janssens-
Maenhout, G., Boesch, H., Pandey, S., Manning, A. J., Jackson, R. B., Nisbet, E. G. and Manning, M. R.: Advancing
Scientific Understanding of the Global Methane Budget in Support of the Paris Agreement, Global Biogeochem. Cycles,
33(12), 1475–1512, doi:https://doi.org/10.1029/2018GB006065, 2019.
Guanter, L., Irakulis-Loitxate, I., Gorroño, J., Sánchez-García, E., Cusworth, D. H., Varon, D. J., Cogliati, S. and Colombo,
R.: Mapping methane point emissions with the PRISMA spaceborne imaging spectrometer, Remote Sens. Environ., 265,
112671, doi:https://doi.org/10.1016/j.rse.2021.112671, 2021.
Hasekamp, O. P. and Butz, A.: Efficient calculation of intensity and polarization spectra in vertically inhomogeneous
scattering and absorbing atmospheres, J. Geophys. Res. Atmos., 113(D20), doi:https://doi.org/10.1029/2008JD010379, 2008.
Hersbach, H., Bell, B., Berrisford, P., Hirahara, S., Horányi, A., Muñoz-Sabater, J., Nicolas, J., Peubey, C., Radu, R.,
Schepers, D., Simmons, A., Soci, C., Abdalla, S., Abellan, X., Balsamo, G., Bechtold, P., Biavati, G., Bidlot, J., Bonavita,
M., De Chiara, G., Dahlgren, P., Dee, D., Diamantakis, M., Dragani, R., Flemming, J., Forbes, R., Fuentes, M., Geer, A.,
Haimberger, L., Healy, S., Hogan, R. J., Hólm, E., Janisková, M., Keeley, S., Laloyaux, P., Lopez, P., Lupu, C., Radnoti, G.,
de Rosnay, P., Rozum, I., Vamborg, F., Villaume, S. and Thépaut, J.-N.: The ERA5 global reanalysis, Q. J. R. Meteorol.
Soc., 146(730), 1999–2049, doi:https://doi.org/10.1002/qj.3803, 2020.
Hoffmann, L., Günther, G., Li, D., Stein, O., Wu, X., Griessbach, S., Heng, Y., Konopka, P., Müller, R., Vogel, B. and
Wright, J. S.: From ERA-Interim to ERA5: the considerable impact of ECMWF's next-generation reanalysis on Lagrangian
transport simulations, Atmos. Chem. Phys., 19(5), 3097–3124, doi:10.5194/acp-19-3097-2019, 2019.
Hu, H., Landgraf, J., Detmers, R., Borsdorff, T., Aan de Brugh, J., Aben, I, Butz, A. and Hasekamp, O.: Toward Global
Mapping of Methane With TROPOMI: First Results and Intersatellite Comparison to GOSAT, Geophys. Res. Lett., 45(8),
3682–3689, doi:https://doi.org/10.1002/2018GL077259, 2018.
Itziar, I.-L., Luis, G., Yin-Nian, L., J., V. D., D., M. J., Yuzhong, Z., Apisada, C., C., W. S., K., T. A., M., D. R., Christian,
F., R., L. D., Benjamin, H., H., C. D., Yongguang, Z., Karl, S., Javier, G., Elena, S.-G., P., S. M., Kaiqin, C., Haijian, Z.,
Jian, L., Xun, L., Ilse, A. and J., J. D.: Satellite-based survey of extreme methane emissions in the Permian basin, Sci. Adv.,
7(27), eabf4507, doi:10.1126/sciadv.abf4507, 2021.
Itziar, I.-L., Luis, G., Yin-Nian, L., J., V. D., D., M. J., Yuzhong, Z., Apisada, C., C., W. S., K., T. A., M., D. R., Christian,
F., R., L. D., Benjamin, H., H., C. D., Yongguang, Z., Karl, S., Javier, G., Elena, S.-G., P., S. M., Kaiqin, C., Haijian, Z.,
Jian, L., Xun, L., Ilse, A. and J., J. D.: Satellite-based survey of extreme methane emissions in the Permian basin, Sci. Adv.,
7(27), eabf4507, doi:10.1126/sciadv.abf4507, 2022.





Jacob, D. J., Turner, A. J., Maasakkers, J. D., Sheng, J., Sun, K., Liu, X., Chance, K., Aben, I., McKeever, J. and
Frankenberg, C.: Satellite observations of atmospheric methane and their value for quantifying methane emissions, Atmos.
Chem. Phys., 16(22), 14371–14396, doi:10.5194/acp-16-14371-2016, 2016.
Jervis, D., McKeever, J., Durak, B. O. A., Sloan, J. J., Gains, D., Varon, D. J., Ramier, A., Strupler, M. and Tarrant, E.: The
GHGSat-D imaging spectrometer, Atmos. Meas. Tech., 14(3), 2127–2140, doi:10.5194/amt-14-2127-2021, 2021.
K., S. J., P., S. D., J., M. M. and V., R.: What Role for Short-Lived Climate Pollutants in Mitigation Policy?, Science (80-. ).,
342(6164), 1323–1324, doi:10.1126/science.1240162, 2013.
Kraut, S., Scharf, L. L. and Butler, R. W.: The adaptive coherence estimator: a uniformly most-powerful-invariant adaptive
detection statistic, IEEE Trans. Signal Process., 53(2), 427–438, doi:10.1109/TSP.2004.840823, 2005.
Lorente, A., Borsdorff, T., Butz, A., Hasekamp, O., aan de Brugh, J., Schneider, A., Wu, L., Hase, F., Kivi, R., Wunch, D.,
Pollard, D. F., Shiomi, K., Deutscher, N. M., Velazco, V. A., Roehl, C. M., Wennberg, P. O., Warneke, T. and Landgraf, J.:
Methane retrieved from TROPOMI: improvement of the data product and validation of the first 2 years of measurements,
Atmos. Meas. Tech., 14(1), 665–684, doi:10.5194/amt-14-665-2021, 2021.
Marchese, A. J., Vaughn, T. L., Zimmerle, D. J., Martinez, D. M., Williams, L. L., Robinson, A. L., Mitchell, A. L.,
Subramanian, R., Tkacik, D. S., Roscioli, J. R. and Herndon, S. C.: Methane Emissions from United States Natural Gas
Gathering and Processing, Environ. Sci. Technol., 49(17), 10718–10727, doi:10.1021/acs.est.5b02275, 2015.
Martin, Van, Damme, Lieven, Clarisse, Simon, Whitburn, Juliette, Hadji-Lazaro and Daniel: Industrial and agricultural
ammonia point sources exposed, Nature, 2018.
Masood, E. and Tollefson, J.: COP26 climate pledges: What scientists think so far, Nature, d41586-021-03034-z,
doi:10.1038/d41586-021-03034-z, 2021.
Mayfield, E. N., Robinson, A. L. and Cohon, J. L.: System-wide and Superemitter Policy Options for the Abatement of
Methane Emissions from the U.S. Natural Gas System, Environ. Sci. Technol., 51(9), 4772–4780,
doi:10.1021/acs.est.6b05052, 2017.
McLinden, C. A., Fioletov, V., Shephard, M. W., Krotkov, N., Li, C., Martin, R. V, Moran, M. D. and Joiner, J.: Space-
based detection of missing sulfur dioxide sources of global air pollution, Nat. Geosci., 9(7), 496–500, 2016.
Mikaloff, F. S. E. and Hinrich, S.: Rising methane: A new climate challenge, Science (80-. )., 364(6444), 932–933,
doi:10.1126/science.aax1828, 2019.
Miller, S. M., Michalak, A. M., Detmers, R. G., Hasekamp, O. P., Bruhwiler, L. M. P. and Schwietzke, S.: China's coal mine
methane regulations have not curbed growing emissions, Nat. Commun., 10(1), 1–8, 2019.
National Oceanic and Atmospheric Administration: Increase in atmospheric methane set another record during 2021,
[online] Available from: https://www.noaa.gov/news-release/increase-in-atmospheric-methane-set-another-record-during-

594    2021, 2022.



Nisbet, E. G., Manning, M. R., Dlugokencky, E. J., Fisher, R. E., Lowry, D., Michel, S. E., Myhre, C. L., Platt, S. M., Allen,
G. and Bousquet, P.: Very Strong Atmospheric Methane Growth in the 4Years 2014–2017: Implications for the Paris
Agreement, Global Biogeochem. Cycles, 33, 2019.
Nisbet, E. G., Fisher, R. E., Lowry, D., France, J. L., Allen, G., Bakkaloglu, S., Broderick, T. J., Cain, M., Coleman, M.,
Fernandez, J., Forster, G., Griffiths, P. T., Iverach, C. P., Kelly, B. F. J., Manning, M. R., Nisbet-Jones, P. B. R., Pyle, J. A.,
Townsend-Small, A., al-Shalaan, A., Warwick, N. and Zazzeri, G.: Methane Mitigation: Methods to Reduce Emissions, on
the Path to the Paris Agreement, Rev. Geophys., 58(1), e2019RG000675, doi:https://doi.org/10.1029/2019RG000675, 2020.
Ocko, I. B., Sun, T., Shindell, D., Oppenheimer, M., Hristov, A. N., Pacala, S. W., Mauzerall, D. L., Xu, Y. and Hamburg, S.
P.: Acting rapidly to deploy readily available methane mitigation measures by sector can immediately slow global warming,
Environ. Res. Lett., 16(5), 54042, doi:10.1088/1748-9326/abf9c8, 2021.
Ouerghi, E., Ehret, T., de Franchis, C., Facciolo, G., Lauvaux, T., Meinhardt, E. and Morel, J.-M.: DETECTION OF
METHANE PLUMES IN HYPERSPECTRAL IMAGES FROM SENTINEL-5P BY COUPLING ANOMALY
DETECTION AND PATTERN RECOGNITION, ISPRS Ann. Photogramm. Remote Sens. Spat. Inf. Sci., V-3–2021, 81–
87, doi:10.5194/isprs-annals-V-3-2021-81-2021, 2021.
Pandey, S., Gautam, R., Houweling, S., Van Der Gon, H. D., Sadavarte, P., Borsdorff, T., Hasekamp, O., Landgraf, J., Tol,
P. and Van Kempen, T.: Satellite observations reveal extreme methane leakage from a natural gas well blowout, Proc. Natl.
Acad. Sci., 116(52), 26376–26381, 2019.
Paoletti, M. E., Haut, J. M., Plaza, J. and Plaza, A.: A new deep convolutional neural network for fast hyperspectral image
classification, ISPRS J. Photogramm. Remote Sens., 145, 120–147, doi:https://doi.org/10.1016/j.isprsjprs.2017.11.021, 2018.
Phil DeCola and WMO Secretariat: An Integrated Global Greenhouse Gas Information System (IG3IS), [online] Available
from: https://public.wmo.int/en/resources/bulletin/integrated-global-greenhouse-gas-information-system-ig3is, 2017.
Reichstein, M., Camps-Valls, G., Stevens, B., Jung, M., Denzler, J., Carvalhais, N. and Prabhat: Deep learning and process
understanding for data-driven Earth system science, Nature, 566(7743), 195–204, doi:10.1038/s41586-019-0912-1, 2019.
Sánchez-García, E., Gorroño, J., Irakulis-Loitxate, I., Varon, D. J. and Guanter, L.: Mapping methane plumes at very high
spatial resolution with the WorldView-3 satellite, Atmos. Meas. Tech. Discuss., 2021, 1–26, doi:10.5194/amt-2021-238,

620 2021.

Saunois, M., Bousquet, P., Poulter, B., Peregon, A., Ciais, P., Canadell, J. G., Dlugokencky, E. J., Etiope, G., Bastviken, D.,
Houweling, S., Janssens-Maenhout, G., Tubiello, F. N., Castaldi, S., Jackson, R. B., Alexe, M., Arora, V. K., Beerling, D. J.,
Bergamaschi, P., Blake, D. R., Brailsford, G., Brovkin, V., Bruhwiler, L., Crevoisier, C., Crill, P., Covey, K., Curry, C.,
Frankenberg, C., Gedney, N., Höglund-Isaksson, L., Ishizawa, M., Ito, A., Joos, F., Kim, H.-S., Kleinen, T., Krummel, P.,
Lamarque, J.-F., Langenfelds, R., Locatelli, R., Machida, T., Maksyutov, S., McDonald, K. C., Marshall, J., Melton, J. R.,
Morino, I., Naik, V., O'Doherty, S., Parmentier, F.-J. W., Patra, P. K., Peng, C., Peng, S., Peters, G. P., Pison, I., Prigent, C.,
Prinn, R., Ramonet, M., Riley, W. J., Saito, M., Santini, M., Schroeder, R., Simpson, I. J., Spahni, R., Steele, P., Takizawa,
A., Thornton, B. F., Tian, H., Tohjima, Y., Viovy, N., Voulgarakis, A., van Weele, M., van der Werf, G. R., Weiss, R.,





Wiedinmyer, C., Wilton, D. J., Wiltshire, A., Worthy, D., Wunch, D., Xu, X., Yoshida, Y., Zhang, B., Zhang, Z. and Zhu,
Q.: The global methane budget 2000–2012, Earth Syst. Sci. Data, 8(2), 697–751, doi:10.5194/essd-8-697-2016, 2016.
Saunois, M., Stavert, A. R., Poulter, B., Bousquet, P., Canadell, J. G., Jackson, R. B., Raymond, P. A., Dlugokencky, E. J.,
Houweling, S., Patra, P. K., Ciais, P., Arora, V. K., Bastviken, D., Bergamaschi, P., Blake, D. R., Brailsford, G., Bruhwiler,
L., Carlson, K. M., Carrol, M., Castaldi, S., Chandra, N., Crevoisier, C., Crill, P. M., Covey, K., Curry, C. L., Etiope, G.,
Frankenberg, C., Gedney, N., Hegglin, M. I., Höglund-Isaksson, L., Hugelius, G., Ishizawa, M., Ito, A., Janssens-Maenhout,
G., Jensen, K. M., Joos, F., Kleinen, T., Krummel, P. B., Langenfelds, R. L., Laruelle, G. G., Liu, L., Machida, T.,
Maksyutov, S., McDonald, K. C., McNorton, J., Miller, P. A., Melton, J. R., Morino, I., Müller, J., Murguia-Flores, F., Naik,
V., Niwa, Y., Noce, S., O'Doherty, S., Parker, R. J., Peng, C., Peng, S., Peters, G. P., Prigent, C., Prinn, R., Ramonet, M.,
Regnier, P., Riley, W. J., Rosentreter, J. A., Segers, A., Simpson, I. J., Shi, H., Smith, S. J., Steele, L. P., Thornton, B. F.,
Tian, H., Tohjima, Y., Tubiello, F. N., Tsuruta, A., Viovy, N., Voulgarakis, A., Weber, T. S., van Weele, M., van der Werf,
G. R., Weiss, R. F., Worthy, D., Wunch, D., Yin, Y., Yoshida, Y., Zhang, W., Zhang, Z., Zhao, Y., Zheng, B., Zhu, Q., Zhu,
Q. and Zhuang, Q.: The Global Methane Budget 2000–2017, Earth Syst. Sci. Data, 12(3), 1561–1623, doi:10.5194/essd-12-

642 1561-2020, 2020.

Schellnhuber, H. J., Rahmstorf, S. and Winkelmann, R.: Why the right climate target was agreed in Paris, Nat. Clim. Chang.,
6(7), 649–653, doi:10.1038/nclimate3013, 2016.
Schurer, A. P., Cowtan, K., Hawkins, E., Mann, M. E., Scott, V. and Tett, S. F. B.: Interpretations of the Paris climate target,
Nat. Geosci., 11(4), 220–221, doi:10.1038/s41561-018-0086-8, 2018.
Sha, M. K., Langerock, B., Blavier, J.-F. L., Blumenstock, T., Borsdorff, T., Buschmann, M., Dehn, A., De Mazière, M.,
Deutscher, N. M., Feist, D. G., García, O. E., Griffith, D. W. T., Grutter, M., Hannigan, J. W., Hase, F., Heikkinen, P.,
Hermans, C., Iraci, L. T., Jeseck, P., Jones, N., Kivi, R., Kumps, N., Landgraf, J., Lorente, A., Mahieu, E., Makarova, M. V,
Mellqvist, J., Metzger, J.-M., Morino, I., Nagahama, T., Notholt, J., Ohyama, H., Ortega, I., Palm, M., Petri, C., Pollard, D.
F., Rettinger, M., Robinson, J., Roche, S., Roehl, C. M., Röhling, A. N., Rousogenous, C., Schneider, M., Shiomi, K., Smale,
D., Stremme, W., Strong, K., Sussmann, R., Té, Y., Uchino, O., Velazco, V. A., Vigouroux, C., Vrekoussis, M., Wang, P.,
Warneke, T., Wizenberg, T., Wunch, D., Yamanouchi, S., Yang, Y. and Zhou, M.: Validation of methane and carbon
monoxide from Sentinel-5 Precursor using TCCON and NDACC-IRWG stations, Atmos. Meas. Tech., 14(9), 6249–6304,
doi:10.5194/amt-14-6249-2021, 2021.
Smith, M. L., Gvakharia, A., Kort, E. A., Sweeney, C., Conley, S. A., Faloona, I., Newberger, T., Schnell, R., Schwietzke, S.
and Wolter, S.: Airborne Quantification of Methane Emissions over the Four Corners Region, Environ. Sci. Technol.,
51(10), 5832–5837, doi:10.1021/acs.est.6b06107, 2017.
Subramanian, R., Williams, L. L., Vaughn, T. L., Zimmerle, D., Roscioli, J. R., Herndon, S. C., Yacovitch, T. I.,
Floerchinger, C., Tkacik, D. S., Mitchell, A. L., Sullivan, M. R., Dallmann, T. R. and Robinson, A. L.: Methane Emissions
from Natural Gas Compressor Stations in the Transmission and Storage Sector: Measurements and Comparisons with the
EPA Greenhouse Gas Reporting Program Protocol, Environ. Sci. Technol., 49(5), 3252–3261, doi:10.1021/es5060258, 2015.



Sun, K., Zhu, L., Cady-Pereira, K., Chan Miller, C., Chance, K., Clarisse, L., Coheur, P.-F., González Abad, G., Huang, G.,
Liu, X., Van Damme, M., Yang, K. and Zondlo, M.: A physics-based approach to oversample multi-satellite, multispecies
observations to a common grid, Atmos. Meas. Tech., 11(12), 6679–6701, doi:10.5194/amt-11-6679-2018, 2018.
T., L., C., G., M., M., A., d'Aspremont, R., D., D., C., D., S. and P., C.: Global assessment of oil and gas methane ultra-
emitters, Science (80-. )., 375(6580), 557–561, doi:10.1126/science.abj4351, 2022.
The European Union's Copernicus Climate Change Service: Annual summary 2021. [online] Available from:
https://climate.copernicus.eu/sites/default/files/custom-uploads/Annual_summary_2021/C3S-CAMS annual temp data and
CO2 2021_press release_final.pdf, n.d.
Thompson, D. R., Thorpe, A. K., Frankenberg, C., Green, R. O., Duren, R., Guanter, L., Hollstein, A., Middleton, E., Ong,
L. and Ungar, S.: Space-based remote imaging spectroscopy of the Aliso Canyon CH4 superemitter, Geophys. Res. Lett.,
43(12), 6571–6578, doi:https://doi.org/10.1002/2016GL069079, 2016.
Thorpe, A. K., Frankenberg, C., Aubrey, A. D., Roberts, D. A., Nottrott, A. A., Rahn, T. A., Sauer, J. A., Dubey, M. K.,
Costigan, K. R., Arata, C., Steffke, A. M., Hills, S., Haselwimmer, C., Charlesworth, D., Funk, C. C., Green, R. O.,
Lundeen, S. R., Boardman, J. W., Eastwood, M. L., Sarture, C. M., Nolte, S. H., Mccubbin, I. B., Thompson, D. R. and
McFadden, J. P.: Mapping methane concentrations from a controlled release experiment using the next generation airborne
visible/infrared imaging spectrometer (AVIRIS-NG), Remote Sens. Environ., 179, 104–115,
doi:https://doi.org/10.1016/j.rse.2016.03.032, 2016.
Turner, A. J., Frankenberg, C. and Kort, E. A.: Interpreting contemporary trends in atmospheric methane, Proc. Natl. Acad.
Sci., 116(8), 2805 LP – 2813, doi:10.1073/pnas.1814297116, 2019.
Tuzson, B., Graf, M., Ravelid, J., Scheidegger, P., Kupferschmid, A., Looser, H., Morales, R. P. and Emmenegger, L.: A
compact QCL spectrometer for mobile, high-precision methane sensing aboard drones, Atmos. Meas. Tech., 13(9), 4715–
4726, doi:10.5194/amt-13-4715-2020, 2020.
United Nations: Global methane pledge, [online] Available from: https://unfccc.int/news/world-leaders-kick-start-
accelerated-climate-action-at-cop26, 2021.
Varon, D., McKeever, J., Jervis, D., Maasakkers, J. D., Pandey, S., Houweling, S., Aben, I., Scarpelli, T. and Jacob, D. J.:
Satellite Discovery of Anomalously Large Methane Point Sources From Oil/Gas Production, Geophys. Res. Lett., 46,
doi:10.1029/2019GL083798, 2019.
Varon, D. J., Jacob, D. J., McKeever, J., Jervis, D., Durak, B. O. A., Xia, Y. and Huang, Y.: Quantifying methane point
sources from fine-scale satellite observations of atmospheric methane plumes, Atmos. Meas. Tech., 11(10), 5673–5686,
doi:10.5194/amt-11-5673-2018, 2018.
Varon, D. J., Jacob, D. J., Jervis, D. and McKeever, J.: Quantifying Time-Averaged Methane Emissions from Individual
Coal Mine Vents with GHGSat-D Satellite Observations, Environ. Sci. Technol., 54(16), 10246–10253,
doi:10.1021/acs.est.0c01213, 2020.



Varon, D. J., Jervis, D., McKeever, J., Spence, I., Gains, D. and Jacob, D. J.: High-frequency monitoring of anomalous
methane point sources with multispectral Sentinel-2 satellite observations, Atmos. Meas. Tech., 14(4), 2771–2785,
doi:10.5194/amt-14-2771-2021, 2021.
Vaughn, T. L., Bell, C. S., Pickering, C. K., Schwietzke, S., Heath, G. A., Pétron, G., Zimmerle, D. J., Schnell, R. C. and
Nummedal, D.: Temporal variability largely explains top-down/bottom-up difference in methane emission estimates from a
natural gas production region, Proc. Natl. Acad. Sci., 115(46), 11712–11717, 2018.
Veefkind, J. P., Aben, I., McMullan, K., Förster, H., de Vries, J., Otter, G., Claas, J., Eskes, H. J., de Haan, J. F., Kleipool,
Q., van Weele, M., Hasekamp, O., Hoogeveen, R., Landgraf, J., Snel, R., Tol, P., Ingmann, P., Voors, R., Kruizinga, B.,
Vink, R., Visser, H. and Levelt, P. F.: TROPOMI on the ESA Sentinel-5 Precursor: A GMES mission for global
observations of the atmospheric composition for climate, air quality and ozone layer applications, Remote Sens. Environ.,
120, 70–83, doi:https://doi.org/10.1016/j.rse.2011.09.027, 2012.
Verhoelst, T., Compernolle, S., Pinardi, G., Lambert, J.-C., Eskes, H. J., Eichmann, K.-U., Fjæraa, A. M., Granville, J.,
Niemeijer, S., Cede, A., Tiefengraber, M., Hendrick, F., Pazmiño, A., Bais, A., Bazureau, A., Boersma, K. F., Bognar, K.,
Dehn, A., Donner, S., Elokhov, A., Gebetsberger, M., Goutail, F., Grutter de la Mora, M., Gruzdev, A., Gratsea, M., Hansen,
G. H., Irie, H., Jepsen, N., Kanaya, Y., Karagkiozidis, D., Kivi, R., Kreher, K., Levelt, P. F., Liu, C., Müller, M., Navarro
Comas, M., Piters, A. J. M., Pommereau, J.-P., Portafaix, T., Prados-Roman, C., Puentedura, O., Querel, R., Remmers, J.,
Richter, A., Rimmer, J., Rivera Cárdenas, C., Saavedra de Miguel, L., Sinyakov, V. P., Stremme, W., Strong, K., Van
Roozendael, M., Veefkind, J. P., Wagner, T., Wittrock, F., Yela González, M. and Zehner, C.: Ground-based validation of
the Copernicus Sentinel-5P TROPOMI NO2 measurements with the NDACC ZSL-DOAS, MAX-DOAS and Pandonia
global networks, Atmos. Meas. Tech., 14(1), 481–510, doi:10.5194/amt-14-481-2021, 2021.
World Meteorological Organization: Global Atmosphere Watch Programme (GAW), [online] Available from:
https://community.wmo.int/activity-areas/gaw, 2022.
Yang, X., Ye, Y., Li, X., Lau, R. Y. K., Zhang, X. and Huang, X.: Hyperspectral Image Classification With Deep Learning
Models, IEEE Trans. Geosci. Remote Sens., 56(9), 5408–5423, doi:10.1109/TGRS.2018.2815613, 2018.
Yu, S., Jia, S. and Xu, C.: Convolutional neural networks for hyperspectral image classification, Neurocomputing, 219, 88–
98, doi:https://doi.org/10.1016/j.neucom.2016.09.010, 2017.
Yuan, Q., Shen, H., Li, T., Li, Z., Li, S., Jiang, Y., Xu, H., Tan, W., Yang, Q., Wang, J., Gao, J. and Zhang, L.: Deep
learning in environmental remote sensing: Achievements and challenges, Remote Sens. Environ., 241, 111716,
doi:https://doi.org/10.1016/j.rse.2020.111716, 2020.
Zavala-Araiza, D., Lyon, D., Alvarez, R. A., Palacios, V., Harriss, R., Lan, X., Talbot, R. and Hamburg, S. P.: Toward a
Functional Definition of Methane Super-Emitters: Application to Natural Gas Production Sites, Environ. Sci. Technol.,
49(13), 8167–8174, doi:10.1021/acs.est.5b00133, 2015.





Zavala-Araiza, D., Alvarez, R. A., Lyon, D. R., Allen, D. T., Marchese, A. J., Zimmerle, D. J. and Hamburg, S. P.: Super-
emitters in natural gas infrastructure are caused by abnormal process conditions, Nat. Commun., 8(1), 14012,
doi:10.1038/ncomms14012, 2017.
Zhang, M., Li, W. and Du, Q.: Diverse Region-Based CNN for Hyperspectral Image Classification, IEEE Trans. Image
Process., 27(6), 2623–2634, doi:10.1109/TIP.2018.2809606, 2018.
