# Peer review of "Toward a versatile spaceborne architecture for immediate"

_Atmospheric Chemistry and Physics, 2022_

## Referee Comment (RC1)

**Review of "**A versatile spaceborne architecture 1 for immediate monitoring of the global methane pledge**"**

**Summary**

This paper proposes an interesting method to address the important issue of quantifying current methane emissions. The authors justifiably argue that no current satellite instrument provides both the coverage and the spatial resolution to accurately measure global methane concentrations; to address this lack they propose a two-step method that uses data from two very different instruments: the wide swath, coarse spatial resolution TROPOMI and narrow swath but very high spatial resolution PRISMA. The TROPOMI data are used to locate high methane emission regions and the methane hotspots within these regions, then the co-located PRISMA data are examined for the presence of plumes. Emissions over the hotspots and plumes are estimated by combining wind speed information with an integrated mass enhancement model.

The approach is demonstrated for short periods over five small regions and the results are compared with surveys over two other regions. The median and range of the plume emissions are qualitatively consistent with those obtained using data from another (non-specified) satellite instrument over the Permian basin, and much higher than those from an aircraft campaign over California. The hotspot and plume emissions are also compared with emissions from the EDGAR_v6.0 inventory; the hotspot emissions were somewhat consistent with the inventory, while the plume emissions were much higher.

Summarizing the above, this is an interesting method with very interesting results. The authors evidently put a great deal of effort and enthusiasm into this work. However, the paper presents several problems, principally lack of detail on how some of the results were obtained. I have listed the main technical issues below, which need to be addressed before the paper can be published. An overarching issue is English language usage. Verb tenses are frequently used incorrectly (e.g, past or conditional future for present), and nouns and adjectives are interchanged. Before resubmitting the authors should have either a native English speaker or someone with excellent English revise the paper. I will be happy to provide more specific wording changes once this been done, if they are still necessary.

1. The method for identifying high emission areas and plumes appears to be visual identification. The authors do mention a Boolean mask for identifying the former, but no details are provided and the reader is left wondering what this means. This needs to be clarified. Such an intensive method is feasible for a small analysis, such as presented in figures 1-3, but obviously not for long term, global emission estimates. Here the authors suggest a machine learning approach for further applications of their method, which is a reasonable suggestion. However, this issue makes the year long results presented in S3 and S4 questionable. Were the TROPOMI maps obtained by applying the Sun oversampling method for an entire year over the original methane

concentrations? If so which wind fields were used to obtain the emissions, both for the regional and plume estimates? How were the PRISMA data averaged over the year? Given the variability in wind direction, I don't think it makes sense to look for plumes in averaged data. These plots need to either explained in much greater detail, or omitted entirely from the paper. If they are to be included, then the authors need to be clear which results (short term or annual) are used in all other plots.

2. The plume maps would be more interesting if the plume source were clearly marked .
3. How was the background vector used in equation 1 derived?
4. What does this sentence mean: methane enhancements detected in spectrometers generally exhibit sparsity, especially over low albedo surfaces.
5. Please define the co-location criteria between the TROPOMI and PRISMA datasets.
6. The section on comparing the TROPOMI/PRISMA results with the California and Permian surveys needs to provide more detail on those surveys (instrument, time of year, temporal and spatial extent). It also needs to emphasize that these comparisons are basically tests of reasonableness, not true quantitative comparisons.
7. The phrase "on a per column basis" is frequently used: what does this mean?
8. The detailed uncertainty analysis is confusing, disorganized and hard to follow. Please put some more thought in how to present this information.

---

## Author Comment (AC1)

**Reply to comments on "Toward a versatile spaceborne architecture for immediate monitoring of the global methane pledge" by Yuchen Wang et al.**

**Reply to Reviewer #1:**

This paper proposes an interesting method to address the important issue of quantifying current methane emissions. The authors justifiably argue that no current satellite instrument provides both the coverage and the spatial resolution to accurately measure global methane concentrations; to address this lack they propose a two-step method that uses data from two very different instruments: the wide swath, coarse spatial resolution TROPOMI and narrow swath but very high spatial resolution PRISMA. The TROPOMI data are used to locate high methane emission regions and the methane hotspots within these regions, then the co-located PRISMA data are examined for the presence of plumes. Emissions over the hotspots and plumes are estimated by combining wind speed information with an integrated mass enhancement model.

The approach is demonstrated for short periods over five small regions and the results are compared with surveys over two other regions. The median and range of the plume emissions are qualitatively consistent with those obtained using data from another (non-specified) satellite instrument over the Permian basin, and much higher than those from an aircraft campaign over California. The hotspot and plume emissions are also compared with emissions from the EDGAR_v6.0 inventory; the hotspot emissions were somewhat consistent with the inventory, while the plume emissions were much higher.

Summarizing the above, this is an interesting method with very interesting results. The authors evidently put a great deal of effort and enthusiasm into this work. However, the paper presents several problems, principally lack of detail on how some of the results were obtained. I have listed the main technical issues below, which need to be addressed before the paper can be published. An overarching issue is English language usage. Verb tenses are frequently used incorrectly (e.g, past or conditional future for present), and nouns and adjectives are interchanged. Before resubmitting the authors should have either a native English speaker or someone with excellent English revise the paper. I will be happy to provide more specific wording changes once this been done, if they are still necessary.

**Response:** We truly appreciate these positive responses and thorough summarizations. We are also very grateful for the valuable comments and suggestions and have addressed all of them in our revised manuscript. Particularly, we have supplemented more technical details to clarify the procedure of our framework. In addition, our co-authors (involving native English speakers) have carefully gone through the entire manuscript to improve the English level.

The followings are our point-to-point responses to the reviewer's comments. The responses are shown in brown font, while the added/rewritten parts are presented in blue font. All revised figures and tables are also included in the manuscripts.

1. The method for identifying high emission areas and plumes appears to be visual identification. The authors do mention a Boolean mask for identifying the former, but no details are provided and the reader is left wondering what this means. This needs to be clarified. Such an intensive method is feasible for a small analysis, such as presented in figures 1-3, but obviously not for long term, global emission estimates. Here the authors suggest a machine learning approach for further applications of their method, which is a reasonable suggestion. However, this issue makes the year long results presented in S3 and S4 questionable. Were the TROPOMI maps obtained by applying the Sun oversampling method for an entire year over the original methane concentrations? If so which wind fields were used to obtain the emissions, both for the regional and plume estimates? How were the PRISMA data averaged over the year? Given the variability in wind direction, I don't think it makes sense to look for plumes in averaged data. These plots need to either explained in much greater detail, or omitted entirely from the paper. If they are to be included, then the authors need to be clear which results (short term or annual) are used in all other plots.

**Response:** Thank you for these valuable comments and suggestions. First, we have supplemented more technical details to clarify the role of the Boolean mask method. As you pointed out, in the first tier of our framework, we apply visual inspection to identify methane hotspots using the TROPOMI-based methane retrievals. The transformation from visual inspection to automatic recognition would significantly advance long-term, global methane monitoring. However, no satisfactory set of criteria is found that could be suitable for this study. This was mainly because, in localized regions, methane budgets respond to the changes in not only super-emitters but also complex external factors (e.g., meteorology, topography, and background concentrations). Similar compromises are also adopted in previous studies. Therefore, automatic recognition enabled by artificial intelligence would play an essential role in the versatile spaceborne architecture for long-term, global methane monitoring (Ouerghi et al., 2021; Paoletti et al., 2018; Yang et al., 2018; Yu et al., 2017; Zhang et al., 2018).

Regarding the identified methane hotspots, we utilize a Boolean mask to select plume-influenced pixels downwind of the source. The background distribution (mean ± standard deviation) is defined by an upwind sample of the measured columns, in which the hourly wind field data came from the ERA5 reanalysis dataset produced by the European Centre for Medium-Range Weather Forecasts (ECMWF) (Hoffmann et al., 2019). We then sample the surrounding ($5 \times 5$) pixels centred on each pixel and compare the corresponding distributions to the background distribution based on a Student's t-test. Pixels with a distribution substantially higher than the background at a confidence level of 95% are assigned to the plume. More details in the Boolean plume mask can be found in previous studies (Pandey et al., 2019; Varon et al., 2018).

Second, we agree that it might make no sense to look for plumes in averaged data due to the variable wind direction and have thus omitted the oversampled methane maps in the first tier of our framework (Fig. S3). In turn, using year-round snapshots in the second tier of our framework, we inspect the identified super-emitters (Figs. 1b ~ 1g) repeatedly and find more methane plumes as expected (Fig. S4). This reinforces the above hypothesis for the widespread occurrence of methane super-emitters.

**Added/rewritten part in Sect. 2.3:** In the first tier of our framework, we apply visual inspection to identify methane hotspots using the TROPOMI-based methane retrievals. The transformation from visual inspection to automatic recognition would significantly advance long-term, global methane monitoring. However, no satisfactory set of criteria is found that could be suitable for this study. This was mainly because, in localized regions, methane budgets respond to the changes in not only super-emitters but also complex external factors (e.g., meteorology, topography, and background concentrations). Similar compromises are also adopted in previous studies. Therefore, automatic recognition enabled by artificial intelligence would play an essential role in the versatile spaceborne architecture for long-term, global methane monitoring (Ouerghi et al., 2021;

Paoletti et al., 2018; Yang et al., 2018; Yu et al., 2017; Zhang et al., 2018).

Regarding the identified methane hotspots, we utilize a Boolean mask to select plume-influenced pixels downwind of the source. The background distribution (mean ± standard deviation) is defined by an upwind sample of the measured columns, in which the hourly wind field data came from the ERA5 reanalysis dataset produced by the European Centre for Medium-Range

Weather Forecasts (ECMWF) (Hoffmann et al., 2019). We then sample the surrounding ($5 \times 5$) pixels centred on each pixel and compare the corresponding distributions to the background distribution based on a Student's t-test. Pixels with a distribution substantially higher than the background at a confidence level of 95% are assigned to the plume. More details in the Boolean plume mask can be found in previous studies (Pandey et al., 2019; Varon et al., 2018).

**Added/rewritten part in Sect. 3.2:** To further explore such a hypothesis, we extend the temporal sample window of our multi-tiered framework. Using year-round snapshots in the second tier of our framework, we inspect the identified super- emitters (Figs. 1b ~ 1g) repeatedly and find more methane plumes as expected (Fig. S3). This reinforces the above hypothesis for the widespread occurrence of methane super-emitters.

2. The plume maps would be more interesting if the plume source were clearly marked.

**Response:** Thanks. We have marked all the plume sources in Fig. 1 and Fig. S3.

3. How was the background vector used in equation 1 derived?

**Response:** Thanks. We have supplemented brief descriptions for this issue. The $\vec{\mu}$ and $\Sigma$ represent the mean background radiance and corresponding covariance, respectively, calculated with their common formulas after subtracting the current signal estimates from the data. Specifically, the $\vec{\mu}$ is calculated from the data with the removal of the most recent enhancement estimates, while the $\Sigma$ is then calculated with updated $\vec{\mu}$ and the most recent enhancement estimates. More technical details are reported in previous studies (Foote et al., 2020). Note that, owing to the non-uniform response of individual detectors in

PRISMA, they are calculated based on per-column spectrums in order to consider different responses of across-track detectors to radiance.

**Added/rewritten part in Sect. 2.2:** The $\vec{\mu}$ and $\Sigma$ represent the mean background radiance and corresponding covariance, respectively, calculated with their common formulas after subtracting the current signal estimates from the data. Specifically, the $\vec{\mu}$ is calculated from the data with the removal of the most recent enhancement estimates, while the $\Sigma$ is then calculated with updated $\vec{\mu}$ and the most recent enhancement estimates. More technical details are reported in previous studies (Foote et al., 2020). Note that, owing to the non-uniform response of individual detectors in PRISMA, they are calculated based on per-column spectrums in order to consider different responses of across-track sensors to radiance.

4. What does this sentence mean: methane enhancements detected in spectrometers generally exhibit sparsity, especially over low albedo surfaces.

**Response:** Sorry for the confusion we caused. We have revised this sentence to clarify this issue. In principle, it would be more difficult to detect methane enhancements in pixels over low-albedo surfaces. Although methane absorption is independent of albedo, the resulting signal in absolute radiance is weakened with surface albedo decreasing.

**Added/rewritten part in Sect. 2.2:** In principle, it would be more difficult to detect methane enhancements in pixels over low-albedo surfaces. Although methane absorption is independent of albedo, the resulting signal in absolute radiance is weakened with surface albedo decreasing.

5. Please define the co-location criteria between the TROPOMI and PRISMA datasets.

**Response:** Thanks. We have supplemented the definition the co-location criteria between the TROPOMI and PRISMA datasets. Regarding the identified regional hotspots, we also apply visual inspection to search for plumes within their surrounding 30 km scales (i.e., corresponding to the swath width of PRISMA) in the second tier of our framework.

**Added/rewritten part in Sect. 2.3:** Regarding the identified regional hotspots, we also apply visual inspection to search for plumes within their surrounding 30 km scales (i.e., corresponding to the swath width of PRISMA) in the second tier of our framework (Irakulis-Loitxate et al., 2021; Lauvaux et al., 2022; Martin et al., 2018; Varon et al., 2020).

6. The section on comparing the TROPOMI/PRISMA results with the California and Permian surveys needs to provide more detail on those surveys (instrument, time of year, temporal and spatial extent). It also needs to emphasize that these comparisons are basically tests of reasonableness, not true quantitative comparisons.

**Response:** Thanks. We have supplemented more technical details on these surveys. The California survey aims to provide the first view of methane super-emitters across the state. This survey is conducted with the Next Generation Airborne Visible/Infrared Imaging Spectrometer (AVIRIS-NG), with 5 nm SWIR spectral sampling, 1.8 km view field, 3 m horizontal resolution, and 3 km cruise altitude, and contains five campaigns over several months from 2016 to 2018. Moreover, this instrument is unique due to its high signal-to-noise ratio and is capable of characterizing methane super-emitters with emissions as small as 2 ~ 10 kg/h for typical surface winds of 5 m/s.

The Permian survey takes advantage of imaging spectroscopy technologies to provide the first spaceborne region-scale and high-resolution survey of methane super-emitters in the Permian basin. This survey is acquired by 30 hyperspectral images from three satellite missions, including Gaofen-5, ZY1, and PRISMA, and focuses on an area of roughly $200 \times 150 \, km^2$ in the Delaware sub-basin of the Permian basin within several days (mostly on four different dates: 15 May 2019, 1 November 2019,

29 December 2019, and 8 February 2020). More technical details on these two surveys can be found in previous studies (Duren et al., 2019; Irakulis-Loitxate et al., 2021).

Moreover, we agree that such comparisons are basically reasonableness test rather than stringently quantitative validations due to measurement divergencies between these datasets (e.g., spatial resolution and detection limit). Collectively, although such comparisons are not quantitative comparisons due to measurement divergencies between these datasets (e.g., spatial resolution and detection limit), they offer further context for the emission magnitude of the identified methane super-emitters and indicate the outstanding strength of our results that could be analogous to abundant outcomes from field campaigns. More importantly, this highlights the urgent need for global monitoring of 'nameless' O&G facilities that possibly emit methane as much as the California field and Permian basin.

**Added/rewritten part in Sect. 3.2:** The California survey aims to provide the first view of methane super-emitters across the state. This survey is conducted with the Next Generation Airborne Visible/Infrared Imaging Spectrometer (AVIRIS-NG), with 5 nm SWIR spectral sampling, 1.8 km view field, 3 m horizontal resolution, and 3 km cruise altitude, and contains five campaigns over several months from 2016 to 2018. Moreover, this instrument is unique due to its high signal-to-noise ratio and is capable of characterizing methane super-emitters with emissions as small as 2 ~ 10 kg/h for typical surface winds of 5

m/s.

The Permian survey takes advantage of imaging spectroscopy technologies to provide the first spaceborne region-scale and high-resolution survey of methane super-emitters in the Permian basin. This survey is acquired by 30 hyperspectral images from three satellite missions, including Gaofen-5, ZY1, and PRISMA, and focuses on an area of roughly $200 \times 150$ km$^2$ in the

Delaware sub-basin of the Permian basin within several days (mostly on four different dates: 15 May 2019, 1 November 2019,

29 December 2019, and 8 February 2020). More technical details on these two surveys can be found in previous studies (Duren et al., 2019; Irakulis-Loitxate et al., 2021).

Collectively, although such comparisons are not quantitative comparisons due to measurement divergencies between these datasets (e.g., spatial resolution and detection limit), they offer further context for the emission magnitude of the identified methane super-emitters and indicate the outstanding strength of our results that could be analogous to abundant outcomes from field campaigns. More importantly, this highlights the urgent need for global monitoring of 'nameless' O&G

facilities that possibly emit methane as much as the California field and Permian basin.

7. The phrase "on a per column basis" is frequently used: what does this mean?

**Response:** Sorry for the confusion we caused. We have supplemented some sentences to explain this phrase at its first appearance. The matched-filter algorithm focuses on the individual columns rather than the whole scene to resolve methane enhancements. This means that the methane enhancement per column is calculated separately (i.e., methane enhancements were calculated on a per-column basis). More explanations can be found in Guanter et al. (2021).

**Added/rewritten part in Sect. 2.2:** The matched-filter algorithm focuses on the individual columns rather than the whole scene to resolve methane enhancements. This means that the methane enhancement per column is calculated separately (i.e., methane enhancements were calculated on a per-column basis). More explanations can be found in Guanter et al. (2021).

8. The detailed uncertainty analysis is confusing, disorganized and hard to follow. Please put some more thought in how to present this information.

**Response:** Thank you very much for this constructive suggestion. We have reorganized and revised the detailed uncertainty analysis in Supplementary Information to clarify this issue, which has been explicitly divided into three sub-issues:

(1) uncertainties in the PRISMA-based methane retrievals; (2) uncertainties in the TROPOMI-based methane emission estimates; and (3) uncertainties in PRISMA-based methane emission estimates. Note that operational TROPOMI-based methane retrieval products have been evaluated strictly and proved to be reliable globally (except in low- and high-albedo and snow-covered areas) (Lorente et al., 2021; Sha et al., 2021) and the related uncertainty analysis is thus omitted here. As a result, we could confirm the reliable performance of our framework. Comprehensive uncertainty analysis is illustrated in

Supplementary Information.

**Reference**

[revised manuscript text omitted]

---

## Author Comment (AC2)

**Reply to comments on "Toward a versatile spaceborne architecture for immediate monitoring of the global methane pledge" by Yuchen Wang et al.**

**Reply to CC #1:**

The article shows a very interesting approach to investigate the different methane emissions using available satellites (TROPOMI and PRIMA) and suggesting that a multitiered constellation could be implemented. Some comments on the article of possible improvements.

**Response:** We truly appreciate your positive responses and valuable comments. We have addressed all of them in our revised manuscript.

The followings are our point-to-point responses to the reviewer's comments. The responses are shown in brown font, while the added/rewritten parts are presented in blue font. All revised figures and tables are also included in the manuscripts.

Line 60 you introduce the term "super-emitters" for first time, the term should be defined better (how big/small, released methane, how spread, etc.) in contrast with hot spots and area sources. This should be tailored for the satellite swath and resolution.

**Response:** Thanks for this valuable comment. We have supplemented the descriptions to clarify the definition of "super-emitters". In this study, super-emitters can generally be defined to be emission sources that comprise highly concentrated methane plumes and dominate localized methane budgets ($\sim 5 \times 5$ km$^2$). In contrast to region-scale hotspots (or area sources), they can be attributed to individual facilities (e.g., factories, chimneys, and pipelines), typically with side lengths varying from several meters to tens of meters depending on monitoring instruments.

**Added/rewritten part in Sect. 1:** Super-emitters can generally be defined to be emission sources that comprise highly concentrated methane plumes and dominate localized methane budgets ($\sim 5 \times 5$ km$^2$). In contrast to region-scale hotspots (or area sources), they can be attributed to individual facilities (e.g., factories, chimneys, and pipelines), typically with side lengths varying from several meters to tens of meters depending on monitoring instruments.

Between lines 80 to 92 a review of existing and capable of detecting methane satellites is shown. However, the swath, passes, resolution, etc. is not given for all satellites. I would suggest to add a table with such information. This would help to better understand/propose a future multi-tiered constellation which could act globally.

**Response:** Thanks. This is a very valuable suggestion. We have supplemented a table (Table 1) to collect the potential satellites and their necessary information (e.g., swath and resolution).

A conclusions section with a better explanation of what number of satellites (which ones in the pipeline / resolution), and
aircrafts needed to have a proper coverage would be needed. Also, would it be night monitoring important, which method or
missions could be used? Atmospheric Lidars? Would the retrieval of structured atmospheric column help the analysis?

**Response:** Very illuminating suggestions. We have supplemented brief discussions to clarify these three issues. Overall,
this multi-tiered framework based on multifarious satellites, aircrafts, and UAVs keeps pursuing wider coverages and faster
revisits. We would thus derive the next objective in this manner, i.e., how to achieve effective, efficient, and economic
monitoring of global methane pledges, in which how to make better coverage-resolution balance between instruments is crucial.
This will be the topic of a next separate study.

Second, yes, nighttime methane monitoring is important because abnormal leakages or pulses might also occur during
nighttime (Plant et al., 2022; Poindexter et al., 2016). In these events, the LIDAR-equipped ones (involving satellites, e.g.,
MERLIN) can allow to retrieve methane fluxes at all-latitudes, all-seasons, and all-weather (involving nighttime) as they are
not relying on sunlight. Fourth, better characterizing methane vertical profile would in principle help to optimize our analysis,
like minimizing the uncertainties in tropospheric air mass factors and subsequent methane enhancements.

**Added/rewritten part in Sect. 3.4:** Note that such a multi-tiered framework based on multifarious satellites, aircrafts,
and UAVs keeps pursuing wider coverages and faster revisits. We would thus derive the next objective in this manner, i.e.,
how to achieve effective, efficient, and economic monitoring of global methane pledges, in which how to make better coverage-
resolution balance between instruments is crucial. This will be the topic of the next separate study.

Third, nighttime methane monitoring is important because abnormal leakages or pulses might also occur during nighttime
(Plant et al., 2022; Poindexter et al., 2016). In these events, the LIDAR-equipped ones (involving satellites, e.g., MERLIN)
can allow to retrieve methane fluxes at all-latitudes, all-seasons, and all-weather (involving nighttime) as they are not relying
on sunlight. Fourth, better characterizing methane vertical profile would help to optimize our analysis, like minimizing the
uncertainties in tropospheric air mass factors and subsequent methane enhancements.

**Cosmetics:**
Spacing between text and references. In Line 57, 59, 136, 223, 225, 244, 312, 343, 360.
**Response:** Thanks. We have supplemented these necessary blank spaces.

Reference in line 117, is this correct format for the current article? In contract to the one in line 145. Is it need to have
same info twice?
**Response:** Thanks. We have checked the format of the reference. Besides, in Line 117 and Line 145, we have deleted the
repetitive references.

**Reference**

Plant, G., Kort, E. A., Brandt, A. R., Chen, Y., Fordice, G., Gorchov Negron, A. M., Schwietzke, S., Smith, M. and Zavala-

Araiza, D.: Inefficient and unlit natural gas flares both emit large quantities of methane, Science (80-. )., 377(6614), 1566–

1571, doi:10.1126/science.abq0385, 2022.

Poindexter, C. M., Baldocchi, D. D., Matthes, J. H., Knox, S. H. and Variano, E. A.: The contribution of an overlooked transport process to a wetland's methane emissions, Geophys. Res. Lett., 43(12), 6276–6284, doi:https://doi.org/10.1002/2016GL068782, 2016.

---

## Author Comment (AC3)

**Reply to comments on "Toward a versatile spaceborne architecture for immediate monitoring of the global methane pledge" by Yuchen Wang et al.**

**Reply to Reviewer #2:**

This paper aims at proposing a framework to utilize current space-borne methane observations to monitor regional emission hotspots and qualify super emitters. The framework combines two methods: one based on global mapping using TROPOMI and the other based on PRISMA (or other high-resolution mappings for small target areas). However, it is not clear what makes this framework different from previous studies (many are cited here), and it is suggested that the authors should clearly state the novel aspects of their method.

**Response:** We truly appreciate this valuable suggestion. We have revised related sentences and supplemented clear statements for the novel aspects of their method. Collectively, existing studies still struggle to surveillance global methane super-emitters due to the fact that individual satellite missions, either TROPOIM or PRISMA, cannot coordinate large-scale swath and high-resolution sampling. To address this issue, we present a two-tiered, space-based framework that coordinates TROPOIM and PRISMA for both planet-scale and plant-level methane retrievals.

**Added/rewritten part in Sect. 1:** Collectively, existing studies still struggle to surveillance global methane super-emitters due to the fact that individual satellite missions, either TROPOIM or PRISMA, cannot coordinate large-scale swath and high-resolution sampling. To address this issue, we present a two-tiered, space-based framework that coordinates TROPOIM and PRISMA for both planet-scale and plant-level methane retrievals.

Additionally, the approach is only demonstrated over short periods for five small areas, and the results are well compared with previous studies. The method for identifying high emission areas and plumes appears to be "visual inspection", which raises questions about how this "framework" could scale to "immediate monitoring of the global methane." This is a key point that needs to be addressed for "a versatile spaceborne architecture." Besides, the detection limit of this method and how it deals with hotspots from natural sources or other anthropogenic sectors other than oil and gas (landfill, agriculture) should be better illustrated before the paper is considered for publication.

**Response:** Thanks for these insightful comments. Yes, we applied visual inspection to identify methane hotspots and plumes using the TROPOMI-based and PRISMA-based methane retrievals. We agree that "visual inspection" is one of the key obstacles to realizing long-term, global methane monitoring. First, we have revised the title to clarify the existing gap to a versatile spaceborne architecture. Second, we have further explained the key role of automatic recognition in long-term, global methane monitoring. The transformation from visual inspection to automatic recognition would significantly advance long-term, global methane monitoring. However, no satisfactory set of automatic criteria is found that could be suitable for this study. This is mainly because, in localized regions, methane budgets respond to the changes in not only super-emitters but also complex external factors (e.g., meteorology, topography, and background concentrations). Similar compromises are also adopted in previous studies. Therefore, automatic recognition enabled by artificial intelligence would play an essential role in the versatile spaceborne architecture for long-term, global methane monitoring (Ouerghi et al., 2021; Paoletti et al., 2018; Yang et al., 2018; Yu et al., 2017; Zhang et al., 2018).

Besides, the detection limit of this framework depends mainly on the TROPOMI-based and PRISMA-based methane retrievals, which have been well discussed in previous studies (Guanter et al., 2021; Hu et al., 2018). Here we have thus supplemented associated discussions on this detection limit briefly. As the robust relationship between the "minimum source" and the related methane enhancement interpreted by Jacob et al. (2016) and Guanter et al. (2021), the detection threshold for the TROPOMI instrument is 4000 kg/h with a wind speed of 5 km/h. Following the same relationship in the PRISMA instrument, we estimate that a retrieval precision of 114 ppb (6.1% with the assumed background concentration of 1850 ppb), such as in the case of the Hassi Messaoud site (Fig. S10e1), would lead to a detection limit of 800 kg/h for the same wind speed (analogous to the reported range of 500 ~ 900 kg/h) (Guanter et al., 2021; Irakulis-Loitxate et al., 2022).

Similar instruments and detection limits are generally comparable to emissions from anthropogenic sectors, like O&G and coal mines in this study or landfills, agriculture, and waste management in previous studies (Maasakkers et al., 2023; Sadavarte et al., 2021; T. et al., 2022). However, no conclusive evidence shows by far that short-term (e.g., daily) satellite-based measurements with such detection limits can capture methane hotspots driven by natural sources (e.g., wetlands). In contrast, long-term (e.g., year-round) satellite-based measurements with much higher detection limits have shown the potential (Pandey et al., 2021).

**Added/rewritten part in Title:** Toward a versatile spaceborne architecture for immediate monitoring of the global methane pledge.

**Added/rewritten part in Sect. 2.3:** The transformation from visual inspection to automatic recognition would significantly advance long-term, global methane monitoring. However, no satisfactory set of criteria is found that could be suitable for this study. This is mainly because, in localized regions, methane budgets respond to the changes in not only super-emitters but also complex external factors (e.g., meteorology, topography, and background concentrations). Similar compromises are also adopted in previous studies. Therefore, automatic recognition enabled by artificial intelligence would play an essential role in the versatile spaceborne architecture for long-term, global methane monitoring (Ouerghi et al., 2021; Paoletti et al., 2018; Yang et al., 2018; Yu et al., 2017; Zhang et al., 2018).

**Added/rewritten part in Sect. 2.5:** The detection limit of this framework depends mainly on the TROPOMI-based and PRISMA-based methane retrievals, which have been well discussed in previous studies (Guanter et al., 2021; Hu et al., 2018). Here we have thus supplemented associated discussions on this detection limit briefly. As the robust relationship between the "minimum source" and the related methane enhancement interpreted by Jacob et al. (2016) and Guanter et al. (2021), the detection threshold for the TROPOMI instrument is 4000 kg/h with a wind speed of 5 km/h. Following the same relationship
in the PRISMA instrument, we estimate that a retrieval precision of 114 ppb (6.1% with the assumed background concentration
of 1850 ppb), such as in the case of the Hassi Messaoud site (Fig. S10e1), would lead to a detection limit of 800 kg/h for the
same wind speed (analogous to the reported range of 500 ~ 900 kg/h) (Guanter et al., 2021; Irakulis-Loitxate et al., 2022).
Similar instruments and detection limits are generally comparable to emissions from anthropogenic sectors, like O&G and
coal mines in this study or landfills, agriculture, and waste management in previous studies (Maasakkers et al., 2023; Sadavarte
et al., 2021; T. et al., 2022). However, no conclusive evidence shows by far that short-term (e.g., daily) satellite-based
measurements with such detection limits can capture methane hotspots driven by natural sources (e.g., wetlands). In contrast,
long-term (e.g., year-round) satellite-based measurements with much higher detection limits have shown the potential (Pandey
et al., 2021).

**Technical Points:**

The title and the abstract are a bit perplexing. The multi-tiered reads mostly two-tiered. I think clarifying these basic
points would be helpful for the reader. In the abstract, it would be nice if the authors could briefly describe what this "versatile
spaceborne architecture" is, and what data it is based on using what methods. At the moment, one needs to read the paper to a
large extent to get some idea of "this framework". The paper could also benefit from adjusting the scope of the text to the
results presented here.

**Response:** Thanks for this constructive suggestion. Accordingly, we have revised the title and abstract to clarify these
key points, particularly distinguishing the two-tiered and versatile spaceborne architectures, and have also adjusted the scope
of the text to the results presented here.

**Added/rewritten part in Title:** Toward a versatile spaceborne architecture for immediate monitoring of the global
methane pledge

**Added/rewritten part in Abstract:** The global methane pledge paves a fresh, critical way toward Carbon Neutrality.
However, it remains largely invisible and highly controversial due to the fact that planet-scale and plant-level methane
retrievals have rarely been coordinated. This has never been more essential within a narrow window to reach the Paris target.
Here we present a two-tiered spaceborne architecture to address this issue. Using this framework, we patrol the world, like the
United States, China, the Middle East, and North Africa, and simultaneously uncover methane-abundant regions and plumes.
These include new super-emitters, potential leakages, and unprecedented multiple plumes in a single source. More importantly,
this framework is shown to challenge official emission reports that possibly mislead estimates from global, regional, to site
scales, particularly by missing super-emitters. Our results show that, in principle, we can extend the above framework to be
multi-tiered by adding upcoming stereoscopic measurements and suitable artificial intelligence, thus versatile for immediate
and future monitoring of the global methane pledge.

Line 51: Ocko et al., 2021 only refers to the anthropogenic methane sources. It is important to state this precisely, not to confuse it with the large portion of methane emissions from natural sources. The current text might be misleading.

**Response:** Sorry for the misleading we caused. We have revised this sentence to make rigorous statements. Fortunately, methane is short-lived ($\sim$ ten years), and, particularly, that from human activities can be reduced in half using existing technologies by 2030 (Ocko et al., 2021).

**Added/rewritten part in Sect. 1:** Fortunately, methane is short-lived ($\sim$ ten years) (J et al., 2013), and, particularly, that from human activities can be reduced in half using existing technologies by 2030 (Ocko et al., 2021).

Line 55, line 59, and many other places: please check references.

**Response:** Thanks. We have carefully gone through the paper to check the references.

Fig. 1 How is "colocation" defined? Using what kind of criteria?

**Response:** Thanks. We have supplemented the definition the co-location criteria between the TROPOMI and PRISMA

datasets. Regarding the identified regional hotspots, we also apply visual inspection to search for plumes within their surrounding 30 km scales (i.e., corresponding to the swath width of PRISMA) in the second tier of our framework (Irakulis-

Loitxate et al., 2021; Martin et al., 2018; T. et al., 2022; Varon et al., 2020).

**Added/rewritten part in Sect. 2.3:** Regarding the identified regional hotspots, we also apply visual inspection to search for plumes within their surrounding 30 km scales (i.e., corresponding to the swath width of PRISMA) in the second tier of our framework (Irakulis-Loitxate et al., 2021; Martin et al., 2018; T. et al., 2022; Varon et al., 2020).

Fig. 2 What temporal periods are considered here to calculate the percentage?

**Response:** Thanks. We have supplemented the description of the temporal periods that are considered to calculate the percentages. The overpass moments are explicitly shown Fig. 1, most of which are inconsistent between for the first- and second-tier monitoring.

**Added/rewritten part in Sect. 3.2:** The overpass moments are explicitly shown Fig. 1, most of which are inconsistent between for the first- and second-tier monitoring.

Fig. 4 How to reconcile PRISMA and TROPOMI results? It seems there are still differences in the order of magnitude.

**Response:** Thanks. Yes, there are differences in the order of magnitude between the TROPOMI-based and PRISMA- based results, and we have supplemented additional discussions to clarify this issue. The main cause is that the TROPOMI- based and PRIMSA-based results represent the methane emissions from different special scales. The former results represent region-scale methane budgets, while the latter ones resolve the emission magnitude from the individual methane super-emitter therein (Fig. 1). Although the latter results can explain a large fraction of the former ones (Fig. 2), the gaps remain mainly due to inconsistent overpass moments between the two-tiered results or sources still missed by the PRIMSA-based results. In other words, closing the temporal gaps between the two tiers or improving the detection ability of the second tier would help to reconcile the first- and second-tiered results.

**Added/rewritten part in Sect. 3.2:** Note that there are differences in the order of magnitude between the TROPOMI-based and PRISMA-based results. The main cause is that the TROPOMI-based and PRIMSA-based results represent the methane emissions from different special scales. The former results represent region-scale methane budgets, while the latter ones resolve the emission magnitude from the individual methane super-emitter therein (Fig. 1). Although the latter results can explain a large fraction of the former ones (Fig. 2), the gaps remain mainly due to inconsistent overpass moments between the two-tiered results or sources still missed by the PRIMSA-based results. In other words, closing the temporal gaps between the two tiers or improving the detection ability of the second tier would help to reconcile the first- and second-tiered results.

---

## Referee Report (RR1)

Second review of "**Toward a  versatile spaceborne architecture for immediate
 monitoring of the global methane pledge**"

The authors have updated their manuscript with suggestions from two reviewers and a reader, and the manuscript is somewhat improved.  I believe it could make a useful contribution to the field of monitoring CH4 from space. But there are still sections that are unclear and many awkward wordings. Below I have listed locations that need some clarification, along with multiple minor corrections.

**Questions:**

Lines 176-178: what does "subtracting the current signal from the data" mean? Which "recent enhancement estimates"?

Line 190 : Is ΔXCH4 multiplied by *f?*

Line 224: Why are the polygons around the plumes masked out? Please clarify.

Line 261: Please summarize the main results of the uncertainty analysis presented in supplement and refer to values shown in Table S1 whenever citing your emission estimates. I assume these uncertainties are used in Figures 3 and 4; please state this clearly.

Lines 301 and 348: Are the Yanquan emissions 30000 kg/h or 7000 kg/h?

Line 311: In line 281 the detection limit for PRISMA is estimated at 800 kg/h. Yet in line 311 the detection threshold is 300 kg/h. Please clarify.

Line 315: The authors state "the overpass timing of TROPOMI can be nearly concordant with that of PRISMA." Ten days does not sound like good co-location. Please justify why ten days is a good enough co-location criteria.

Line 337: The authors state: "To this end, we apply a multi-spectral retrieval algorithm to eliminate this effect to a large extent. The detailed illustrations are shown in Supplementary Information (Fig. S5)." Please provide a sentence or two on the algorithm used.

Line 401: Please explain what is meant by "spatial proxies".

Line 405: A compromise between what? Maybe the authors mean a combination of inventory data and downwind measurements?

Line 409: Are the authors stating that the Rumaila and Hassi Messaud EDGAR emissions are biased low with respect to he results in this paper? Please make this clearer. Please explain why the factors in this paragraph would apply only to these two locations.

Line 485: Does the shading in the violin plots represent the uncertainty In each plume estimate? Or something else? Please clarify.

**Minor editing suggestions:**

Replace **multi-tiered** with **two-tiered** wherever the current work is discussed.

Line 34: within "the" narrow window…

Line 35: We focused on several regions (United States, China, the Middle East,
36 and North Africa,) and …

Line 36: and uncovered …

Line 40: and thus is sufficiently versatile for ….

Line 46: within the narrow window …

Line 49: it has been rising since 2007, with a surge in 2014 and a record high in 2021 (insert references  I omitted)

Line 53: policymakers

Line 54: on the eve of the Paris target, large uncertainties in emissions remain, and thus hinder …

Line 60:  for example, field campaigns report nearly double official claims of methane emissions in the United States by detecting missing leaks

Line 65: defined as emission sources that …

Line 68: with dimensions varying from …

Line 72: In contrast to area sources (e.g., cities), super-emitters are typically coal mines, wells, gathering stations, storage tanks, pipelines, and flares, with diameters on the order of dozens of metres or less, but generating plums of highly concentrated methane.

Line 80: spatially limited

Line 81: and miss many super emitters

Line 87: wide swaths and high-resolution sampling have not been simultaneously available

Line 88: Recently global methane monitoring has become possible..

Line 90: It provides daily global methane columns,

Line 91: and a high signal-to-noise ratio

Line 92: Next-generation satellite missions, pioneered by the GHGSat constellation (three satellites at the moment), have emerged

Line 96: great potential

Line 98: Note that the regions these satellites usually observe are  already know to contain many super-emitters

Line 101: existing studies still struggle to survey global methane super-emitters due to the fact that individual satellite missions, such as TROPOMI or PRISMA, do not both have a wide swath and high resolution sampling.

Line 103: TROPOMI

Line 106: Using this framework, we focused on China, the United States, Iraq, Kuwait, and Algeria

Line 107: We also monitored a single source to map multiple plumes and to look for possible methane leaks.

Line 109: is not in place, the two-tiered satellite constellation presented in this
 study has great potential for measuring progress towards global methane pledges

Line 114: due to its large swath (~2600 km)

Line 115: revisit time, moderate footprint …, and excellent sounding precision and accuracy.

Line 116: TROPOMI observes approximately

Line 117: the first consisting of near infrared

Line 127: super-emitters due to their unprecedented resolution

Line 138: Two-tiered methane retrievals

Line 139: we employ the operational TROPOMI methane products.

Line 140: which is retrieved

Line 160: especially for observations from instruments deployed on satellite and aircraft

Line 163: can implicitly account for

Line 168: the physically based method requires background concentrations that are …

Line 173: The calculation process of methane enhancements ($\Delta$XCH4, ppb) is as follows.

Line 179: in PRISMA, enhancements are calculated …

Line 186: with decreasing surface albedo

Line 197: Two-tiered attribution

Line 203: in a versatile spaceborne …

Line 218: progressively decreasing downwind

Line 222: and originate from …

Line 233:  in high source regions, such as megacities, there are likely super-emitters that are undetectable following our method.

Line 235: Two-tiered quantification

Line 257: these processes have been described in previous studies

Line 259: the $U10$ term., which typically has a random error on the order of 50%

Line 263: that can monitor global methane pledges

Line 265: originates.  We need to account for

Line 278: As the robust relationship between the "minimum source" and the related methane enhancement developed by Jacob et al. (2016) and Guanter et al. (2021) shows, the detection threshold for the TROPOMI instrument is

Line 280: for the PRISMA instrument …

Line 287: shown potential for monitoring natural methane hotspots

Line 307: plumes originate

Line 338: the only explanation

Line 339: This has previously only been seen in …. Therefore, our multi-tiered outcomes indicate there are more widespread methane leaks than have been previously detected. Note that the multi-spectral retrieval algorithm cannot completely remove the albedo effects on our results. However, our methods could lead to targeted on-site re-inspection on O&G fields worldwide.

Line 343: Our framework detects

Line 346: current satellite constellations alone

Line 347: More satellites could capture changes during even shorter time windows.

Line 349: Figure 2 illustrates the extent to which the second-tier of our two-tiered satellite constellation explains the regional budget detected by the first tier.

Line 350: Delete this sentence: The overpass times (in Fig. 1) are usually different between the first and second tier observations.

Line 351. The share of the regional budget due to the plumes ranges from 8.2% (Hassi Messaud)  to 53.8 ~ 65.9% (Rumaila, Burgan, and Wattenberg).

Line 354: different overpass time.

Line 361: this reinforces our hypothesis that

Line 363: different spatial scales

Line 366: different overpass times between the two-tiered results

Line 369: A regional survey in a California field provides some useful data for evaluating our results, owing to

Lines 371: The survey was conducted

Line 373: and included five campaigns

Line 375: The survey reports 1181 methane plumes, more than 500 times the number of plumes reported by previous aerial studies.

Line 377: Even though some regions of interest in our study are far less well known  than the California fields,

Line 378: the plumes detected by

Line 380: were conducted

Line 380: Satellite observations taken over the Permian basin ((one of the top O&G bases worldwide) from 2019 to 2020 (need reference here) provide additional comparison data.

Line 381: took advantage of

Line 383: survey acquired

Line 387: basin reported a much higher number of strong methane super-emitters, whose median emission rates (1850 kg/h) are much closer to

Line 388: although such comparisons are not quantitative due to many differences in measurement characteristics (e.g., spatial resolution and detection limit),

Line 389: they provide context for the emission magnitudes of the methane super-emitters we have identified and indicate  that our results are within the range of values obtained from field campaigns.

Line 391: More importantly, these results highlight

Line 392: possibly emit as much methane as the California fields and Permian basin.

Line 393: Comparing emissions from our two-tiered approach with a state of the art methane emission inventory (EDGARv6.0) for 2018, (Fig. 4), we find that our emission estimates using TROPOMI data over methane hotspots are roughly consistent with the inventory, with biases ranging from -49.9% to +91.8% with an average bias of 63.2%.  The exception is the Hassi Messaoud field in Algeria where the O&G sector is in rapid development: here our estimate is 498.2% of the EDGARv6.0 inventory.

Line 398: On the other hand, our estimates using PRISMA data over plumes are orders of magnitude greater than the EDGARv6.0 emissions. This suggests that traditional emission inventories may have acceptable performance for methane abundant regions but may grossly underestimate emission from methane super-emitters.

Line 401: There are a number of possible explanations for the low estimates from EDGARv6.0

Line 421: We have presented a two-tiered …

Line 422: We have demonstrated this framework with examples from around the world, with synergistic …

Line 423: We have located new methane super-emitters, tracked potential methane leakages from storage tanks, and resolved multiple methane plumes from a single source.

Line 426: our results suggest inventories miss unknown super-emitters and underestimate emission magnitudes, partly due to a surge in the number of oil and gas (O&G) facilities and widespread abnormalities in O&G operations.

Line 428: Our data prove that existing satellite missions can already lead to immediate …

Line 429: While window for achieving the Paris target is rapidly closing, our approach can provide improved methane emission estimates before the deployment of more advanced instruments, which can also be integrated into our system.

Line 432: Delete sentence starting with "In addition .."

Line 435: It should be noted that the multi-tiered framework is extremely flexible.

Line 441: based on multiple satellites, aircrafts, and UAVs will provide greater spatial coverages and more frequent revisits

Line 442: This flexibility will provide effective, efficient, and economic monitoring of global methane pledges, though this will require careful balancing of coverage and resolution between instruments.

Line 444: of our next study.

Line 445: LIDAR instruments (e.g., MERLIN (need reference) can retrieve methane fluxes day and night at all latitudes, in all-seasons, and in all-weather.

Line 447: Fourth, better characterizing methane vertical profiles would help to optimize our analysis, by minimizing the uncertainties in tropospheric air mass factors and subsequent methane enhancements.

Line 448: Finally, rapid advances in artificial intelligence (AI) techniques can significantly speed up the detection of faint signals from methane enhancements, and to …

Line 456: Still, large gaps remain in coverage and implementation (?). This is especially true for low- and middle-income countries, where tight budgets dim the hopes for filling these gaps by 2030, while methane emissions are likely to rise as countries continue to develop. In this context, the present framework can serve as a cost-effective component of the global methane monitoring network and thus support fair climate negotiations between countries.

This framework harmonizes global scale and high-resolution methane retrievals, with a dual focus on mapping region-scale and plant-level drivers. In this work the framework reconciles the wide swath of TROPOMI (i.e., ~ 2600 km) with the high resolution of PRISMA (i.e., 30x30 m2), in contrast to conventional satellite-based surveys, which suffer from either low resolution or narrow swaths.. Looking forward, developments of Earth's monitoring platforms (e.g., satellites, aircrafts, and unmanned drones) and artificial intelligence will continue to strengthen the performance of methane plume retrievals and emission estimates. On eve of the Paris target, at least while a methane product obtained from a instrument with a wide swath, high resolution, and agile analysis is not in place, our multi-tiered satellite constellation has important implications for measuring global methane pledges.

Line 464: Methane-abundant regions and associated super-emitters as captured by TROPOMI and PRISMA locations are marked by black rectangles and dots. Placenames were obtained from GoogleMaps, and are usually the names of the nearest O&G fields and coal mines. (b ~ g) Each row presents a methane-abundant region and the super-emitters detected within it (b1 ~ b4, c1 ~ c4, d1 ~ d4, e1 ~ e4, f1 ~ f2, and g1 ~ g2). For each super emitter (five-pointed stars), the overpass times of the multi-tiered satellite constellation and the consequent emissions estimate are presented. The base maps were obtained from GoogleMaps. The second color bar for PRISMA images is suitable for the super-emitters in China, while the first applies for other countries. Plume sources in the PRISMA results are marked by red circles.

Line 483: shown in Fig. 1. The 1:1 line is shown by grey dashes.

Line 486: The images of TROPOMI, MethaneSAT, PRISMA, and EnMAP are obtained from http://www.tropomi.eu/, https://www.methanesat.org, https://www.asi.it/en/earth-science/prisma/, and https://www.enmap.org/, respectively. The methane maps from TROPOMI and PRISMA refer to the results in Figs. 1e and 1b1. The grey marks indicate upcoming platforms (i.e., MethaneSAT and EnMAP) and techniques (e.g., AI techniques that can optimize the identification and quantification ofmethane super-emitters).

---

## Author Response (AR2)

**Reply to comments on "Toward a versatile spaceborne architecture for immediate monitoring of the global methane pledge" by Yuchen Wang et al.**

**Reply to Reviewer #1:**

The authors have updated their manuscript with suggestions from two reviewers and a reader, and the manuscript is somewhat improved. I believe it could make a useful contribution to the field of monitoring $CH_4$ from space. But there are still sections that are unclear and many awkward wordings. Below I have listed locations that need some clarification, along with multiple minor corrections.

**Response:** Thank you for your valuable feedback on our manuscript. Accordingly, we have addressed all of the comments provided by the reviewer, particularly revising the sections requiring clarification and the awkward phrasings meticulously. We are confident that these modifications have enhanced the manuscript's clarity and contribution to the field of monitoring $CH_4$ from space. Thank you again for your time and expertise in providing us with this constructive feedback.

**Questions:**

Lines 176-178: what does "subtracting the current signal from the data" mean? Which "recent enhancement estimates"?

**Response:** Thanks. We have revised these sentences to clarify these statements (i.e., "the current signal from the data" and "recent enhancement estimates").

The $\vec{\mu}$ and $\Sigma$ represent the mean value and covariance of the background radiance, respectively. To avoid any contamination of the target spectrum into these background parameters, we estimate them with an iterative approach by removing all gas enhancement signals. More technical details are reported in previous studies (Foote et al., 2020).

**Added/rewritten part in Sect. 2.2:** The $\vec{\mu}$ and $\Sigma$ represent the mean value and covariance of the background radiance, respectively. To avoid any contamination of the target spectrum into these background parameters, we estimate them with an iterative approach by removing all gas enhancement signals. More technical details are reported in previous studies (Foote et al., 2020).

Line 190: Is $\Delta \mathbf{XCH_4}$ multiplied by $f$?

**Response:** Yes, $\Delta XCH_4$ is multiplied by $f$. We have revised this sentence to clarify this issue. $\Delta XCH_4$ is then scaled by this pixel-specific scalar ($f$) and thus normalized by the albedo term, similar to the per-pixel normalization in previous hyperspectral analysis (Kraut et al., 2005).

**Added/rewritten part in Sect. 2.2:** $\Delta XCH_4$ is then scaled by this pixel-specific scalar ($f$) and thus normalized by the albedo term, similar to the per-pixel normalization in previous hyperspectral analysis (Kraut et al., 2005).

Line 224: Why are the polygons around the plumes masked out? Please clarify.

**Response:** Apologies for this confusion that may have been caused by the unintentionally misleading information provided. We have revised this sentence in which the word "out" is redundant.

**Added/rewritten part in Sect. 2.2:** Finally, we manually draw polygons to mask such resulting plumes.

Please summarize the main results of the uncertainty analysis presented in supplement and refer to values shown in

Table S1 whenever citing your emission estimates. I assume these uncertainties are used in Figures 3 and 4; please state this clearly.

**Response:** Thanks. We have supplemented the main results of the uncertainty analysis here, in which appropriate references are made to Supplementary Information. As demonstrated in Supplementary Information, our comprehensive uncertainty analysis establishes the robustness of our estimates, with uncertainties being entirely controllable within a range of -70% (Table S1). Such uncertainties are also used and shown in Figs. 1 ~ 4.

**Added/rewritten part in Sect. 2.4:** As demonstrated in the Supplementary Information, our comprehensive uncertainty analysis establishes the robustness of our estimates, with uncertainties being entirely controllable within a range of -70%

(Table S1). Such uncertainties are shown in Figs. 1 ~ 4.

Lines 301 and 348: Are the Yanquan emissions 30000 kg/h or 7000 kg/h?

**Response:** Thanks. The emission rate of 30000 kg/h was detected from the hotspot (i.e., the Yangquan field) via the

TROPOMI-based monitoring, while the emission rate of 7,000 kg/h was detected from the plume (i.e., a specific coal mine in the Yangquan field) via the PRISMA-based monitoring. We have thus revised the associated sentences to clarify the statements.

**Added/rewritten part in Sect. 3.1:** Besides the well-known oil fields (Figs. 1c ~ 1f), methane hotspots have also emerged in developing coal mine fields such as the Yangquan field, which exhibit comparable emission levels (> 30000

kg/h) (Fig. 1g).

**Added/rewritten part in Sect. 3.2:** Fourth, a distinct methane plume appears in a coal mine in a mountainous area (in the Yangquan field, China), exceeding all of the detected O&G super-emitters regarding the emission rate (> 7000 kg/h)

(Fig. 1g1).

Line 311: In line 281 the detection limit for PRISMA is estimated at 800 kg/h. Yet in line 311 the detection threshold is

300 kg/h. Please clarify.

**Response:** Apologies for this confusion. We have corrected this inconsistence.

**Added/rewritten part in Sect. 3.1:** Such precise distinctions benefit from the high resolution of the second-tiered monitoring, despite being limited by the relatively higher detection threshold (~ 800 kg/h).

Line 315: The authors state "the overpass timing of TROPOMI can be nearly concordant with that of PRISMA." Ten days does not sound like good co-location. Please justify why ten days is a good enough co-location criteria.

**Response:** We agree with the reviewer that there are no hard co-location criteria by far. We have thus deleted this statements and revised the preceding sentence to clarify this issue. For a given set (including both a methane-abundant region and associated super-emitters), the overpass timing of TROPOMI can be nearly concordant with that of PRISMA in some cases.

**Added/rewritten part in Sect. 3.1:** For a given set (including both a methane-abundant region and associated super- emitters), the overpass timing of TROPOMI can be nearly concordant with that of PRISMA in some cases.

Line 337: The authors state: "To this end, we apply a multi-spectral retrieval algorithm to eliminate this effect to a large extent. The detailed illustrations are shown in Supplementary Information (Fig. S5)." Please provide a sentence or two on the algorithm used.

**Response:** Thanks. We have supplemented a brief description for this algorithm. To this end, we apply a multi-spectral retrieval algorithm to eliminate this effect to a large extent. We utilize two spectral bands to launch the matched-filtered algorithm separately: one that is highly sensitive to methane absorption (i.e., 2300 nm) and another that is much less sensitive (i.e., 1700 nm) but exhibit similar surface and aerosol reflectance properties. Figure S5 shows that the 2300 nm- driven matched-filtered algorithm result in noticeable methane vestiges above the storage tanks, while the 1700 nm-driven algorithm does not.

**Added/rewritten part in Sect. 3.2:** To this end, we apply a multi-spectral retrieval algorithm to eliminate this effect to a large extent. We utilize two spectral bands to launch the matched-filtered algorithm separately: one that is highly sensitive to methane absorption (i.e., 2300 nm) and another that is much less sensitive (i.e., 1700 nm) but exhibit similar surface and aerosol reflectance properties. Figure S5 shows that the 2300 nm-driven matched-filtered algorithm result in noticeable methane vestiges above the storage tanks, while the 1700 nm-driven algorithm does not.

Line 401: Please explain what is meant by "spatial proxies"

**Response:** Thanks. We have supplemented a brief description for "spatial proxies". To establish bottom-up methane emission inventories, we need to allocate area sources to regular grids based on spatial information, like nighttime lights (so- called spatial proxies).

**Added/rewritten part in Sect. 3.3:** To establish bottom-up methane emission inventories, we need to allocate area sources to regular grids based on spatial information, like nighttime lights (so-called spatial proxies) (Geng et al., 2017).

Line 405: A compromise between what? Maybe the authors mean a combination of inventory data and downwind measurements?

**Response:** Thanks. We have revised this sentence to clarify this statement. Generally, because of technical difficulties or safety risks, we have to compromise to measure such abnormal emissions downwind rather than on sites.

**Added/rewritten part in Sect. 3.3:** Generally, because of technical difficulties or safety risks, we have to compromise to measure such abnormal emissions downwind rather than on sites.

Line 409: Are the authors stating that the Rumaila and Hassi Messaud EDGAR emissions are biased low with respect to the results in this paper? Please make this clearer. Please explain why the factors in this paragraph would apply only to these two locations.

**Response:** Yes. The emissions of the super-emitters in the Rumaila and Hassi Messaud fields in the EDGARv6.0 are still less than our results (Fig. 1b2, and Fig. 1e3, and Fig. 4), and such factors might also apply to other sources. Thus, we have revised these sentences to clarify these statements.

**Added/rewritten part in Sect. 3.3:** Third, the above divergence between our plant-based estimates and the EDGARv6.0 might also be explained by other causes such as outdated emission factors.

Line 485: Does the shading in the violin plots represent the uncertainty In each plume estimate? Or something else? Please clarify.

**Response:** Thanks. The shading represents the number distribution of the methane plumes with different emission rates. We have supplemented a brief description to clarify this issue.

**Added/rewritten part in Fig. 4:** The shading represents the number distribution of the methane plumes with different emission rates.

**Minor editing suggestions:**

*Response: We are very grateful to the reviewer for such meticulous editing suggestions. We have adopted all the suggestions.*

Replace multi-tiered with two-tiered wherever the current work is discussed.

**Response:** Thanks. We have replaced the "multi-tiered" with "two-tiered" in the full manuscript.

Line 34: within "the" narrow window…

**Response:** Thanks. We have completed the revision accordingly.

Line 35: We focused on several regions (United States, China, the Middle East, and North Africa,) and …

**Response:** Thanks. We have completed the revision accordingly.

Line 36: and uncovered …

**Response:** Thanks. We have completed the revision accordingly.

Line 40: and thus is sufficiently versatile for ….

**Response:** Thanks. We have completed the revision accordingly.

Line 46: within the narrow window….

**Response:** Thanks. We have completed the revision accordingly.

Line 49: it has been rising since 2007, with a surge in 2014 and a record high in 2021 (insert references I omitted)

**Response:** Thanks. We have completed the revision accordingly.

Line 53: policymakers

**Response:** Thanks. We have completed the revision accordingly.

Line 54: on the eve of the Paris target, large uncertainties in emissions remain, and thus hinder …

**Response:** Thanks. We have completed the revision accordingly.

Line 60: for example, field campaigns report nearly double official claims of methane emissions in the United States by
detecting missing leaks.

**Response:** Thanks. We have completed the revision accordingly.

Line 65: defined as emission sources that

**Response:** Thanks. We have completed the revision accordingly.

Line 68: with dimensions varying from …

**Response:** Thanks. We have completed the revision accordingly.

Line 72: In contrast to area sources (e.g., cities), super-emitters are typically coal mines, wells, gathering stations, storage tanks, pipelines, and flares, with diameters on the order of dozens of metres or less, but generating plums of highly concentrated methane.

**Response:** Thanks. We have completed the revision accordingly.

Line 80: spatially limited

**Response:** Thanks. We have completed the revision accordingly.

Line 81: and miss many super emitters

**Response:** Thanks. We have completed the revision accordingly.

Line 87: wide swaths and high-resolution sampling have not been simultaneously available

**Response:** Thanks. We have completed the revision accordingly.

Line 88: Recently global methane monitoring has become possible.

**Response:** Thanks. We have completed the revision accordingly.

Line 90: It provides daily global methane columns,

**Response:** Thanks. We have completed the revision accordingly.

Line 91: and a high signal-to-noise ratio

**Response:** Thanks. We have completed the revision accordingly.

Line 92: Next-generation satellite missions, pioneered by the GHGSat constellation (three satellites at the moment), have emerged

**Response:** Thanks. We have completed the revision accordingly.

Line 96: great potential

**Response:** Thanks. We have completed the revision accordingly.

Line 98: Note that the regions these satellites usually observe are already know to contain many super-emitters

**Response:** Thanks. We have completed the revision accordingly.

Line 101: existing studies still struggle to survey global methane super-emitters due to the fact that individual satellite
missions, such as TROPOMI or PRISMA, do not both have a wide swath and high resolution sampling.
**Response:** Thanks. We have completed the revision accordingly.
Line 103: TROPOMI
**Response:** Thanks. We have completed the revision accordingly.
Line 106: Using this framework, we focused on China, the United States, Iraq, Kuwait, and Algeria
**Response:** Thanks. We have completed the revision accordingly.
Line 107: We also monitored a single source to map multiple plumes and to look for possible methane leaks.
**Response:** Thanks. We have completed the revision accordingly.
Line 109: is not in place, the two-tiered satellite constellation presented in this study has great potential for measuring
progress towards global methane pledges
**Response:** Thanks. We have completed the revision accordingly.
Line 114: due to its large swath (~2600 km)
**Response:** Thanks. We have completed the revision accordingly.
Line 115: revisit time, moderate footprint …, and excellent sounding precision and accuracy.
**Response:** Thanks. We have completed the revision accordingly.
Line 116: TROPOMI observes approximately
**Response:** Thanks. We have completed the revision accordingly.
Line 117: the first consisting of near infrared
**Response:** Thanks. We have completed the revision accordingly.
Line 127: super-emitters due to their unprecedented resolution
**Response:** Thanks. We have completed the revision accordingly.
Line 138: Two-tiered methane retrievals
**Response:** Thanks. We have completed the revision accordingly.

Line 139: we employ the operational TROPOMI methane products.

**Response:** Thanks. We have completed the revision accordingly.

Line 140: which is retrieved

**Response:** Thanks. We have completed the revision accordingly.

Line 160: especially for observations from instruments deployed on satellite and aircraft

**Response:** Thanks. We have completed the revision accordingly.

Line 163: can implicitly account for

**Response:** Thanks. We have completed the revision accordingly.

Line 168: the physically based method requires background concentrations that are …

**Response:** Thanks. We have completed the revision accordingly.

Line 173: The calculation process of methane enhancements ($\Delta XCH4$, ppb) is as follows.

**Response:** Thanks. We have completed the revision accordingly.

Line 179: in PRISMA, enhancements are calculated …

**Response:** Thanks. We have completed the revision accordingly.

Line 186: with decreasing surface albedo

**Response:** Thanks. We have completed the revision accordingly.

Line 197: Two-tiered attribution

**Response:** Thanks. We have completed the revision accordingly.

Line 203: in a versatile spaceborne …

**Response:** Thanks. We have completed the revision accordingly.

Line 218: progressively decreasing downwind

**Response:** Thanks. We have completed the revision accordingly.

Line 222: and originate from …

**Response:** Thanks. We have completed the revision accordingly.

Line 233: in high source regions, such as megacities, there are likely super-emitters that are undetectable following our method.

**Response:** Thanks. We have completed the revision accordingly.

Line 235: Two-tiered quantification

**Response:** Thanks. We have completed the revision accordingly.

Line 257: these processes have been described in previous studies

**Response:** Thanks. We have completed the revision accordingly.

Line 259: the $U10$ term., which typically has a random error on the order of 50%

**Response:** Thanks. We have completed the revision accordingly.

Line 263: that can monitor global methane pledges

**Response:** Thanks. We have completed the revision accordingly.

Line 265: originates. We need to account for

**Response:** Thanks. We have completed the revision accordingly.

Line 278: As the robust relationship between the "minimum source" and the related methane enhancement developed by Jacob et al. (2016) and Guanter et al. (2021) shows, the detection threshold for the TROPOMI instrument is

**Response:** Thanks. We have completed the revision accordingly.

Line 280: for the PRISMA instrument …

**Response:** Thanks. We have completed the revision accordingly.

Line 287: shown potential for monitoring natural methane hotspots

**Response:** Thanks. We have completed the revision accordingly.

Line 307: plumes originate

**Response:** Thanks. We have completed the revision accordingly.

Line 338: the only explanation

**Response:** Thanks. We have completed the revision accordingly.

Line 339: This has previously only been seen in …. Therefore, our multi-tiered outcomes indicate there are more widespread methane leaks than have been previously detected. Note that the multi-spectral retrieval algorithm cannot completely remove the albedo effects on our results. However, our methods could lead to targeted on-site re-inspection on O&G fields worldwide.

**Response:** Thanks. We have completed the revision accordingly.

Line 343: Our framework detects

**Response:** Thanks. We have completed the revision accordingly.

Line 346: current satellite constellations alone

**Response:** Thanks. We have completed the revision accordingly.

Line 347: More satellites could capture changes during even shorter time windows.

**Response:** Thanks. We have completed the revision accordingly.

Line 349: Figure 2 illustrates the extent to which the second-tier of our two-tiered satellite constellation explains the regional budget detected by the first tier.

**Response:** Thanks. We have completed the revision accordingly.

Line 350: Delete this sentence: The overpass times (in Fig. 1) are usually different between the first and second tier observations.

**Response:** Thanks. We have completed the revision accordingly.

Line 351. The share of the regional budget due to the plumes ranges from 8.2% (Hassi Messaud) to 53.8 ~ 65.9% (Rumaila, Burgan, and Wattenberg).

**Response:** Thanks. We have completed the revision accordingly.

Line 354: different overpass time.

**Response:** Thanks. We have completed the revision accordingly.

Line 361: this reinforces our hypothesis that

**Response:** Thanks. We have completed the revision accordingly.

Line 363: different spatial scales

**Response:** Thanks. We have completed the revision accordingly.

Line 366: different overpass times between the two-tiered results

**Response:** Thanks. We have completed the revision accordingly.

Line 369: A regional survey in a California field provides some useful data for evaluating our results, owing to

**Response:** Thanks. We have completed the revision accordingly.

Lines 371: The survey was conducted

**Response:** Thanks. We have completed the revision accordingly.

Line 373: and included five campaigns

**Response:** Thanks. We have completed the revision accordingly.

Line 375: The survey reports 1181 methane plumes, more than 500 times the number of plumes reported by previous aerial studies.

**Response:** Thanks. We have completed the revision accordingly.

Line 377: Even though some regions of interest in our study are far less well known than the California fields,

**Response:** Thanks. We have completed the revision accordingly.

Line 378: the plumes detected by

**Response:** Thanks. We have completed the revision accordingly.

Line 380: were conducted

**Response:** Thanks. We have completed the revision accordingly.

Line 380: Satellite observations taken over the Permian basin (one of the top O&G bases worldwide) from 2019 to 2020

(need reference here) provide additional comparison data.

**Response:** Thanks. We have completed the revision accordingly.

Line 381: took advantage of

**Response:** Thanks. We have completed the revision accordingly.

Line 383: survey acquired

**Response:** Thanks. We have completed the revision accordingly.

Line 387: basin reported a much higher number of strong methane super-emitters, whose median emission rates (1850 kg/h) are much closer to

**Response:** Thanks. We have completed the revision accordingly.

Line 388: although such comparisons are not quantitative due to many differences in measurement characteristics (e.g., spatial resolution and detection limit),

**Response:** Thanks. We have completed the revision accordingly.

Line 389: they provide context for the emission magnitudes of the methane super-emitters we have identified and indicate that our results are within the range of values obtained from field campaigns.

**Response:** Thanks. We have completed the revision accordingly.

Line 391: More importantly, these results highlight

**Response:** Thanks. We have completed the revision accordingly.

Line 392: possibly emit as much methane as the California fields and Permian basin.

**Response:** Thanks. We have completed the revision accordingly.

Line 393: Comparing emissions from our two-tiered approach with a state of the art methane emission inventory (EDGARv6.0) for 2018, (Fig. 4), we find that our emission estimates using TROPOMI data over methane hotspots are roughly consistent with the inventory, with biases ranging from -49.9% to +91.8% with an average bias of 63.2%. The exception is the Hassi Messaoud field in Algeria where the O&G sector is in rapid development: here our estimate is 498.2% of the EDGARv6.0 inventory.

**Response:** Thanks. We have completed the revision accordingly.

Line 398: On the other hand, our estimates using PRISMA data over plumes are orders of magnitude greater than the EDGARv6.0 emissions. This suggests that traditional emission inventories may have acceptable performance for methane abundant regions but may grossly underestimate emission from methane super-emitters.

**Response:** Thanks. We have completed the revision accordingly.

Line 401: There are a number of possible explanations for the low estimates from EDGARv6.0

**Response:** Thanks. We have completed the revision accordingly.

Line 421: We have presented a two-tiered …

**Response:** Thanks. We have completed the revision accordingly.

Line 422: We have demonstrated this framework with examples from around the world, with synergistic …

**Response:** Thanks. We have completed the revision accordingly.

Line 423: We have located new methane super-emitters, tracked potential methane leakages from storage tanks, and resolved multiple methane plumes from a single source.

**Response:** Thanks. We have completed the revision accordingly.

Line 426: our results suggest inventories miss unknown super-emitters and underestimate emission magnitudes, partly due to a surge in the number of oil and gas (O&G) facilities and widespread abnormalities in O&G operations.

**Response:** Thanks. We have completed the revision accordingly.

Line 428: Our data prove that existing satellite missions can already lead to immediate …

**Response:** Thanks. We have completed the revision accordingly.

Line 429: While window for achieving the Paris target is rapidly closing, our approach can provide improved methane emission estimates before the deployment of more advanced instruments, which can also be integrated into our system.

**Response:** Thanks. We have completed the revision accordingly.

Line 432: Delete sentence starting with "In addition .."

**Response:** Thanks. We have completed the revision accordingly.

Line 435: It should be noted that the multi-tiered framework is extremely flexible.

**Response:** Thanks. We have completed the revision accordingly.

Line 441: based on multiple satellites, aircrafts, and UAVs will provide greater spatial coverages and more frequent revisits

**Response:** Thanks. We have completed the revision accordingly.

Line 442: This flexibility will provide effective, efficient, and economic monitoring of global methane pledges, though this will require careful balancing of coverage and resolution between instruments.

**Response:** Thanks. We have completed the revision accordingly.

Line 444: of our next study.

**Response:** Thanks. We have completed the revision accordingly.

Line 445: LIDAR instruments (e.g., MERLIN (need reference) can retrieve methane fluxes day and night at all latitudes, in all-seasons, and in all-weather.

**Response:** Thanks. We have completed the revision accordingly.

Line 447: Fourth, better characterizing methane vertical profiles would help to optimize our analysis, by minimizing the uncertainties in tropospheric air mass factors and subsequent methane enhancements.

**Response:** Thanks. We have completed the revision accordingly.

Line 448: Finally, rapid advances in artificial intelligence (AI) techniques can significantly speed up the detection of faint signals from methane enhancements, and to …

**Response:** Thanks. We have completed the revision accordingly.

Line 456: Still, large gaps remain in coverage and implementation (?). This is especially true for low- and middle- income countries, where tight budgets dim the hopes for filling these gaps by 2030, while methane emissions are likely to rise as countries continue to develop. In this context, the present framework can serve as a cost-effective component of the global methane monitoring network and thus support fair climate negotiations between countries. This framework harmonizes global scale and high-resolution methane retrievals, with a dual focus on mapping region-scale and plant-level drivers. In this work the framework reconciles the wide swath of TROPOMI (i.e., ~ 2600 km) with the high resolution of

PRISMA (i.e., 30x30 m2), in contrast to conventional satellite-based surveys, which suffer from either low resolution or narrow swaths.. Looking forward, developments of Earth's monitoring platforms (e.g., satellites, aircrafts, and unmanned drones) and artificial intelligence will continue to strengthen the performance of methane plume retrievals and emission estimates. On eve of the Paris target, at least while a methane product obtained from an instrument with a wide swath, high resolution, and agile analysis is not in place, our multi-tiered satellite constellation has important implications for measuring global methane pledges.

**Response:** Thanks. We have completed the revision accordingly.

Line 464: Methane-abundant regions and associated super-emitters as captured by TROPOMI and PRISMA locations are marked by black rectangles and dots. Placenames were obtained from GoogleMaps, and are usually the names of the nearest O&G fields and coal mines. (b ~ g) Each row presents a methane-abundant region and the super-emitters detected within it (b1 ~ b4, c1 ~ c4, d1 ~ d4, e1 ~ e4, f1 ~ f2, and g1 ~ g2). For each super emitter (five-pointed stars), the overpass times of the multi-tiered satellite constellation and the consequent emissions estimate are presented. The base maps were obtained from GoogleMaps. The second color bar for PRISMA images is suitable for the super-emitters in China, while the first applies for other countries. Plume sources in the PRISMA results are marked by red circles.

**Response:** Thanks. We have completed the revision accordingly.

Line 483: shown in Fig. 1. The 1:1 line is shown by grey dashes.

**Response:** Thanks. We have completed the revision accordingly.

Line 486: The images of TROPOMI, MethaneSAT, PRISMA, and EnMAP are obtained from http://www.tropomi.eu/, https://www.methanesat.org, https://www.asi.it/en/earthscience/prisma/, and https://www.enmap.org/, respectively. The methane maps from TROPOMI and PRISMA refer to the results in Figs. 1e and 1b1. The grey marks indicate upcoming platforms (i.e., MethaneSAT and EnMAP) and techniques (e.g., AI techniques that can optimize the identification and quantification ofmethane super-emitters)

**Response:** Thanks. We have completed the revision accordingly.